# THE QUEST FOR WINNING TICKETS IN LOW-RANK ADAPTERS

## ABSTRACT

Low-Rank Adaptation (LoRA), a prominent parameter-efficient fine-tuning (PEFT) method, offers an effective strategy for adapting large pre-trained models to specific tasks with minimal computational overhead. LoRA achieves this by introducing low-rank parameter matrices to the frozen pre-trained models. However, despite their efficiency, LoRA and its variants modify all elements of a parameter block, which is unnecessary as LoRA primarily aims to adjust a small set of subspaces that capture task-specific knowledge. Drawing inspiration from the Lottery Ticket Hypothesis (LTH), which posits that dense neural networks contain sparse sub-networks capable of performing similarly to fully-parameterized models, we investigate whether similar sparse subnetworks exist for low-rank adapters. We demonstrate that such subnetworks, often referred to as "winning tickets" in the context of LTH, indeed exist for low-rank adapters. We introduce a method to identify this sparse subset of weights for each layer by relating the top subspaces of the pretrained parameter block to the elements of the corresponding weight matrix. This subset is then fine-tuned using LoRA. We show that this sparse subset is not necessarily unique; as long as sparsity is kept within a certain bound defined by the task, random subnetworks with similar sparsity can act as winning tickets. Building on this discovery, we propose a novel approach called Partial-LoRA, which adds sparse low-rank parameters to pre-trained models. Through extensive experiments on 8 vision and 4 language tasks, we demonstrate that Partial-LoRA can reduce trainable parameters by up to 87% while maintaining or even improving model performance in some cases. Our work thus reduces memory needs and theoretically grounds sparse LoRAs.

## 1 INTRODUCTION

Parameter-efficient fine-tuning (PEFT) methods (Xin et al., 2024) have emerged as a compelling approach to adapt large pre-trained models to specific tasks without the computational overhead of training all model parameters. These methods are crucial for efficiency, reduced energy consumption, lower costs, deployment on resource-limited devices, and allow for democratization of large foundation models. These methods are in contrast to traditional fine-tuning approaches which often involve adding special-purpose layers (Mengde Xu, 2023), adjusting normalization (Giannou et al., 2023), retraining, or introducing parallel layers (Chen et al., 2022). Low-rank adaptation (LoRA) (Hu et al., 2022), as the most prominent PEFT approach, strikes a middle ground between efficiency and performance by adjusting a subset of weight subspaces using low-rank matrices, thereby intending to impact only a small portion of the information embedded in a model.

Despite these advancements, fine-tuning using LoRAs still involves modifying every element of the original weight matrix, even when employing low-rank residual matrices. However, large pre-trained models have low intrinsic dimensionality, allowing for effective finetuning with fewer parameters (Aghajanyan et al., 2020; Hu et al., 2022). This, combined with the lottery ticket hypothesis (LTH) (Frankle & Carbin, 2019), suggests smaller sub-networks ('winning tickets') may achieve similar performance, i.e., for fine-tuning, only a small subset of weights in each layer needs modification.

Aiming to find these smaller sub-networks, we extend the theoretical work on LTH and random masking (Gadhikar et al., 2023) to low-rank adaptation. We show that for each layer, randomly masking the weights of a low-rank adapter leads to a sufficiently tight bound on the residuals added

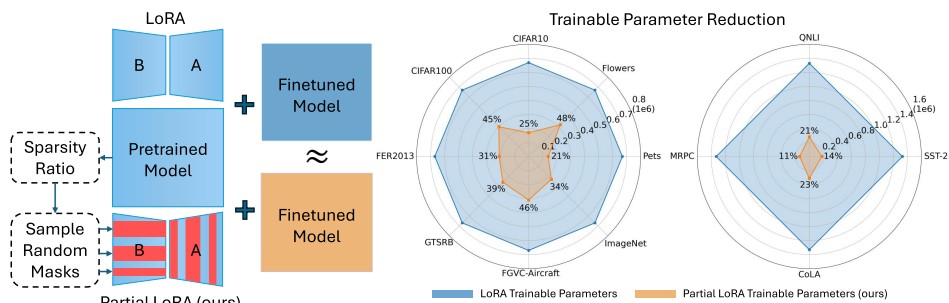

Figure 1: Process of sparsifying LoRAs is shown on the left. We extract sparsity ratios from the pretrained model and use them to sample random masks for the components of the low-rank adapter. This leads to outputs similar to the fully-parameterized LoRA while significantly lowering the number of trainable parameters as shown on the right for both vision and language tasks.

on top of the outputs of the layer through the adapter. This mask's sparsity may depend on the model's required capacity to perform accurately on a downstream task. This way, we show LoRA residuals may be significantly sparsified while maintaining performance, similar to random pruning of full models (Chijiwa et al., 2022), thereby delivering a winning ticket. These subnetworks are therefore not unique and as long as the sparsity is kept within the identified bound, any randomly masked LoRA bearing the same sparsity can obtain similar results to a standard LoRA.

Motivated by these theoretical findings, we propose a method to identify the per-layer sparsity ratio required for a given downstream task. With our proposed approach, we aim to relate the top subspaces of the pre-trained weights to the specific elements of the weight matrix and extract the sparsity ratio for that weight matrix. This allows us to limit fine-tuning of the pre-trained weights to these sub-networks using our approach which we name Partial-LoRA. This process is visualized in Figure 1. We also explore the available methods in the literature of pruning (Lee et al., 2019; Molchanov et al., 2019) to assign importance values to the elements of the pre-trained weight matrix as sparsity ratio derivation alternatives and show the similarity between our approach and these methods. We also demonstrate that our approach can extract the sparsity ratio without the need to compute the output gradients with respect to the weights.

Our experiments with visual and language tasks demonstrate that random masking using our approach reduces the number of parameters needed for residuals by up to 87% while maintaining performance in most cases and even noticeably improving accuracy in some instances in the visual domain. We compare results from Partial-LoRA, where low-rank residuals are randomly sparsified, to cases where specific subnetworks are targeted through deterministic pruning approaches. This way, we show that Partial-LoRA obtains a better performance on average compared to targeting specific subnetworks. This confirms that the precise selection of specific weights might be less critical if the overall quantity of trainable parameters (sparsity factor) is maintained within strategic limits, which we determine for each layer using the importance values derived through the pretrained model.

We also perform ablation studies on the rank and the magnitude of the residuals to study the behavior of masked LoRAs. Moreover, we extend our experiments to compare few-shot training to training on the entire dataset. We find that masked LoRAs maintain similar performance to LoRAs, indicating robustness across dataset sizes. We also explore a shift from deterministic to stochastic methods in our sub-network extraction, treating importance scores as probabilistic to sample parameters to show the minimal impact of this sampling procedure. Our main contributions are as follows.

- We provide a theoretical ground for the existence of lottery tickets in LoRAs, demonstrating that randomly masked LoRAs can achieve performance on par with fully parameterized LoRAs across a wide range of datasets and modalities. This shows that even with significant sparsification, the performance can be maintained, highlighting the robustness of our approach.

- We show that principled sub-network selection is not strictly necessary. Given access to a sufficient number of elements, LoRAs can effectively adjust the key subspaces of the pre-trained weight matrix, ensuring efficient fine-tuning in LoRAs. This holds for commonly used importance measures from the pruning literature and the proposed SVD-based method targeting the top subspaces of the pretrained weights.

- Our experiments demonstrate that random masking of LoRAs can allow for up to a 70% reduction in the number of parameters on ImageNet1k and up to 87% in the case of MRPC with minimal changes in performance, even outperforming LoRAs in some instances. This effectively reduces the number of trainable parameters from 110 million in the case of full-rank fine-tuning of ViT-B-16 to 120k, allowing fine-tuning without specialized hardware, further democratizing personalization of large models. This substantial decrease in parameter use, combined with robust performance across varied dataset sizes and LoRA ranks, validates the effectiveness of random masking strategies.

## 2 RELATED WORK

Parameter Efficient Fine-Tuning (PEFT) methods adapt large, overparameterized models with fewer trainable parameters than full retraining requires. A commonly used PEFT approach, namely Low-Rank Adaptation (LoRA) (Hu et al., 2022), uses trainable low-rank matrices added to each layer, significantly reducing the training parameter count. AdaLoRA (Zhang et al., 2023c) refines LoRAs by adaptively modifying the rank of the residual matrices during training, enhancing the performance of LoRAs. Conversely, DoRA (Liu et al., 2024) and LoRA+ (Hayou et al., 2024) address discrepancies between low-rank and full-rank fine-tuning, pointing out limitations in LoRA's training dynamics. Our method builds on LoRAs and AdaLoRAs but could also complement other LoRA-based advancements, potentially matching the dynamics of full-rank fine-tuning and achieving similar model optimization.

Building on the foundation set by LoRAs, works such as LoRAPrune (Zhang et al., 2023b) and LoRA-Shear (Chen et al., 2023) propose approaches to model pruning. LoRAPrune introduces a LoRA-guided pruning method that utilizes the weights and gradients of LoRA adaptors rather than pre-trained weights for importance estimation, coupled with a structured iterative pruning process. LoRAShear employs a structured pruning strategy that initially creates dependency graphs over LoRA modules to identify minimally removable structures. It should be noted that both methods deal with pruning the original model and do not focus on masking LoRAs. Alternatively, aiming to tailor the rank value in a layer-specific manner for the task at hand, PRILoRA (Benedek & Wolf, 2024) linearly allocates a different rank for the residuals at each layer. While this approach obtains some improvements in performance, there is no guarantee for reduction of the number of parameters. Additionally, the same rank value can be used alongside our approach to tailor the masked LoRAs for each layer given the same layer-rank schedule.

More in line with the parameter reduction perspective of our work, IncreLoRA (Zhang et al., 2023a) separates the components of the adapter into multiple rank-1 modules that add up to the full residual, enabling the addition of more modules based on the task requirements. Moreover, there are upperbounds implemented for these additions to prevent overparamterization. Unlike IncreLoRA our approach focuses on using randomized masking within the existing LoRA parameters. While IncreLoRA seeks to enhance parameter efficiency by dynamically adjusting ranks to better fit task-specific demands, our method strategically masks parameters for layers based on the requirements of the task, thus maintaining more streamlined implementation and lower computational costs. (Kopiczko et al., 2024) uses a pair of frozen low-rank matrices shared across layers and adapts them with trainable scaling vectors, demonstrating similar performance to LoRA while reducing trainable parameters. However, the overall parameters are increased during training depending on the size and number of weight matrices involved. A more directly comparable work would be (Xu & Zhang, 2024), where LoRA residuals are randomly masked out and gradients are not applied to the masked parameters. These random masks are generated at the start of training based on a predetermined sparsity. This is in contrast to our work where we use the pretrained model to determine the sparsity required for each layer, making our approach adaptive based on the task's required capacity in a principled manner. Additionally, while approaches like PEFT-Masking (Xu & Zhang, 2024), LoRAPrune, LoRA-Shear, PRILoRA, IncreaLoRA and VeRA improve parameter efficiency or adaptively distribute ranks, they lack theoretical bounds for maintaining approximation quality. Our work bridges this gap by providing a formal theoretical bound on the difference between outputs of masked and unmasked LoRAs.

From a theoretical perspective, (Gadhikar et al., 2023)–building up on the previous work from (Burkholz et al., 2022; Burkholz, 2022)–proved the existence of strong lottery tickets in randomly masked variants of large overparameterized models. This is done by building on subset sum approxi-

mation (Lueker, 1998) (a computational approach that addresses the challenge of selecting a subset of numbers that sum up to a specific target). While previous proofs required the original model to be comprised of double the layers of the target model, (Gadhikar et al., 2023) used a one-for-one layer approach where only $L + 1$ layers are required from the original model for the target (winning ticket) model to be approximated through pruning. In this work, we extend this proof to show that the original formulation still holds in the domain of low-rank adaptation.

## 3 BACKGROUND

Given a model $M$ parameterized by $W, b$ with depth $L$, LoRA (Hu et al., 2022) fine-tunes the weight matrix $\mathbf{W}^l \in \mathbb{R}^{m \times n}$ at layer $l$ by introducing a low-rank residual trainable matrix $\Delta \mathbf{W}_l \in \mathbb{R}^{m \times n}$. This residual is added to the weight matrix $\mathbf{W}_l$ of the frozen pre-trained model $M$. While no bias terms are added to the layer, the only trainable parameters of the fine-tuning process are the residuals added at each layer. This results in each layer of the new model being formulated as:

$$\mathbf{h}_l = \sigma\left((\mathbf{W}_l + \Delta \mathbf{W}_l)\mathbf{x} + \mathbf{b}_l\right), \tag{1}$$

where $\mathbf{h}_l$ is the output of the layer and $\mathbf{b}_l$ represents the bias term at layer $l$ from the original pre-trained model. $\sigma$ and $\mathbf{x}$ represent the layer nonlinearity and input, respectively. The matrix $\Delta \mathbf{W}_l$ itself is defined as:

$$\Delta \mathbf{W}_l = \mathbf{B}_l \mathbf{A}_l, \ \ \mathbf{B}_l \in \mathbb{R}^{m \times d}, \ \mathbf{A}_l \in \mathbb{R}^{d \times n}, \tag{2}$$

where $\mathbf{B}_l$ and $\mathbf{A}_l$ are two low-rank trainable matrices. Using this formulation of the residual, the number of trainable parameters is reduced from $m \times n$ to $m \times d + n \times d$ where $d \ll \min(m, n)$.

## 4 METHODOLOGY

Here, we outline our work divided into two key subsections. section 4.1 extends the theoretical framework of Strong Lottery Tickets (SLT) to Low-Rank Adapters (LoRAs). We demonstrate that assuming a target low-rank adapter (the winning ticket) performing well on a given task exists, a pretrained model finetuned using a randomly masked LoRAs can approximate this target LoRA if the unmasked adapter is wider by a logarithmic margin. In section 4.2 we describe the process of extracting subnetworks (masks for LoRAs) from pretrained models using a small number of labeled instances. This involves determining the importance of each element in the weight matrix of the pretrained model and selectively pruning them from the adapter while maintaining accuracy.

### 4.1 EXISTENCE OF LOTTERY TICKETS IN LoRA RESIDUALS

In this section, we extend the existing theoretical work on Strong Lottery Tickets (SLT) (Gadhikar et al., 2023) to show that a randomly pruned LoRA can approximate a target LoRA (winning ticket), provided the unpruned LoRA is wider by a logarithmic margin. In other words, the pruned LoRA represents a subnetwork of the fully parameterized one, capable of approximating a target network without requiring the initial overparameterized model.

**Theorem 4.1** *Define a network $f_T$ of depth $L$, parameterized by pretrained weights and biases $W_l$ and $b_l$, with low-rank adapters at each layer parameterized by residuals $\Delta W_T^l$. Additionally, define a pruned network $f_{LoRA}$ of depth $L + 1$, parameterized by $\Delta W^l \cdot U$, where $U \sim B(p^l)$ is a mask sampled from a Bernoulli distribution, and the same $W_l$, $b_l$ as the target model. This pruned model consists of sparsity factors $p_l$ at layer $l$, and the residuals $\Delta W_{ij}^l \sim U([-1, 1])$. Both networks have $n_{T,l}$ and $n_{LoRA,l}$ neurons at layer $l$. Then, given variables $\epsilon, \delta \in (0, 1)$, with failure probability $1 - \delta$, there exists a mask $U$ such that, for all $x \in \mathcal{D}$ within the compact space $\mathcal{D}$, it holds that $\max_{x \in \mathcal{D}} ||f_T(x; \Delta W_T) - f_{LoRA}(x; \Delta W \cdot U)|| \leq \epsilon$ if:*

$$n_{LoRA,l} \geq C \frac{n_{T,l}}{\log(1/1 - p_{l+1})} \log\left(\frac{1}{\min(\epsilon_l, \delta/\rho)}\right). \tag{3}$$

Where $C$ is a distribution dependent constant and $\rho$ and $\epsilon_l$ are defined in Appendix A following (Gadhikar et al., 2023; Burkholz, 2022). The proof is provided in Appendix A and hinges on the idea that after multiplying the neurons in the target network by a certain factor, we can ensure that at least

one edge remains unpruned after pruning the original model, preserving the network's functionality. We show the changes to each layer made by the trainable residuals of LoRAs does not alter the proof in (Gadhikar et al., 2023; Burkholz, 2022).

A concern with this masking approach is flow preservation (Burkholz, 2022). In some randomly pruned models, two consecutive layers can end up with mismatched pruned weights, causing any input to those layers to result in a zero output due to a discontinuity. This happens when non-zero outputs from the first layer are multiplied by pruned elements in the second layer. To address this, a common modification is to adjust the non-zero indices to maintain input flow. However, in this work, this step is unnecessary because the pretrained model's frozen layers remain unchanged, ensuring that flow is preserved even if the low-rank residual weights are set to zero.

Theorem 4.1 shows that given a full-parameter adapter, there exists a randomly masked adapter dependent on layer width capable of delivering performance that matches the full-parameter adapter while using fewer parameters. In practice, we show that while the exact ratio of this reduction in the number of parameters is unknown beforehand (since the target LoRA is unknown), importance measuring criterion from the pruning literature may be used as a proxy to obtain this ratio. We name this ratio the capacity at each layer required to learn the task at hand.

### 4.2 FINDING SUBNETWORKS IN PRETRAINED MODELS

To extract sparsity ratios from pretrained models, we propose a two-step approach. First, we present an algorithm for deriving sparsity ratios based on the model's performance on a small subset of labeled instances. Then, we discuss various importance measures that can be used to rank the significance of elements in the weight matrix. This section details both the algorithmic approach (section 4.2.1) and the importance measures (section 4.2.2) used in our method.

#### 4.2.1 DERIVING SPARSITY RATIOS

We start by randomly sampling a small subset of labeled instances from the dataset in a few-shot scenario. This subset, $\mathcal{D}_t$, with $m$ samples from the dataset $\mathcal{D}$, is used to measure the pre-trained model's accuracy as a baseline. Importance measures are then employed to rank the significance of each element in the weight matrix. The process for importance measures is detailed in section 4.2.2. Starting with the least important elements, weights are progressively masked until the model's accuracy drops below 90% of the baseline, using the same few-shot dataset $\mathcal{D}_t$. This iterative masking is applied to the pre-trained model, yielding a sparsity ratio for each layer, as outlined in Algorithm 1. Random masks are then generated using these ratios by sampling from a Bernoulli distribution. During low-rank finetuning, only the unmasked elements of weight matrices at each layer are modified.

---

**Algorithm 1** Sparsity ratio derivation for a single layer

---

Pretrained model $M$, weight matrix $\mathbf{W}^l$ for layer $l$ of $M$, input-output data pairs $(\mathbf{x}, \mathbf{y})$ forming dataset $D_t$, where $D_t \subset D$ consists of $m$ shots randomly sampled per class from $D$

1: $\mu \leftarrow \frac{1}{N} \sum_{(x,y) \in D_t} \mathbb{I}[M(\mathbf{x}, \mathbf{W}) = y]$          ▷ Determine few-shot accuracy on $D_t$
2: $\hat{\mathbf{y}} \leftarrow M(\mathbf{x})$          ▷ Compute model output for $\mathbf{x} \in D_t$
3: $I^l \leftarrow \text{importance}(\hat{\mathbf{y}}, \mathbf{y}, \mathbf{W}^l)$       ▷ Compute importance scores for every element of $\mathbf{W}^l$
4: $\mathbf{U} \leftarrow \mathbf{1}$          ▷ Initialize mask matrix with ones
5: **while** $0.9 \times \mu \leq \frac{1}{N} \sum_{(\mathbf{x},y) \in D_t} \mathbb{I}[M(\mathbf{x}, \mathbf{W}^l \odot \mathbf{U}) = y]$ **do**
6:      $I^l \leftarrow I^l \setminus \text{argmin}(I^l)$          ▷ Remove least important index from $I^l$
7:      $\mathbf{U}_{ij} \leftarrow 0$ for all $(i,j) \notin I^l$      ▷ Update mask matrix to zero out less important weights
8: **end while**
9: **return** $|I^l|$

---

#### 4.2.2 IMPORTANCE MEASURES

Commonly used importance measures (Lee et al., 2019; Molchanov et al., 2019) focus on an element-level computation deriving a scalar value for each element in a weight matrix. However, based on the formulation of low-rank adaptation detailed in section 3, it is enough to infer the important

rows and columns of the weight matrix and there is no necessity to focus on specific elements when dealing with LoRAs. This is because for every element of the residual we have $\Delta w_{ij} = \sum_k^d \mathbf{b}_{i,k} \mathbf{a}_{k,j}$. Therefore, rather than masking the residual weights $\Delta \mathbf{W} = \mathbf{BA}$ we can opt to instead mask $\mathbf{B}$ and $\mathbf{A}$ separately which removes any requirement for computing a large matrix that will be masked before the forward pass of the inputs. This implies that the unimportant rows of $\mathbf{W}$ will lead to masking rows of $\mathbf{B}$ while unimportant columns will lead to masking $\mathbf{A}$. To this end, we compute the singular value decomposition of the weight matrix $\mathbf{W} = \mathbf{P\Lambda Q}$ where $\mathbf{P}$, $\mathbf{Q}$, and $\mathbf{\Lambda}$ represent the left and right singular vectors and the singular values, respectively. Then, we use the similarity of the rows or columns of the matrix with the right singular vectors as a proxy for importance. We start by deriving the weighted right singular vectors of the pretrained weight matrix as follows:

$$\mathbf{V} = \mathbf{Q\Lambda} \,, \tag{4}$$

Then, to derive the importance values for rows of $\mathbf{W}^l$, the following equation is used:

$$I_{SVD}^l = ||\mathbf{W}^l \mathbf{V}||_2, \tag{5}$$

Where $I_{SVD}^l$ represents the importance of each row index. The same process may be repeated for $\mathbf{W}^T$ to infer the column importance values. $I_{SVD}^l$ can then be used in Algorithm 1 to extract the importance values. Alternatively, any importance measure in the pruning literature such as SNIP (Lee et al., 2019) or IMP (Molchanov et al., 2019) may be used. SNIP, as a single-shot pruning approach, uses the absolute value of the gradient of the output with respect to the weight matrix at every layer as a proxy for the importance which makes the method dependent on the pairs of samples passed through the network. IMP, on the other hand, scales the gradients by the weights to also consider the magnitude of weights alongside their gradient. Both of these approaches compute the importance in an element-wise manner. Therefore, to derive the indices required for adaptation, we can take the row-wise or column-wise sum over the importance values. Regardless of the importance criterion, to derive the mask $\mathbf{U}$ in Algorithm 1, we sample row and column masks $\mathbf{u}_{row}$ and $\mathbf{u}_{col}$ according to $|I^l|$ obtained from Algorithm 1 where:

$$\Delta \mathbf{W} = (\mathbf{u}_{row} \cdot \mathbf{B})(\mathbf{u}_{col} \cdot \mathbf{A}) \,. \tag{6}$$

Here, $\mathbf{u}_{row} \sim \text{Bernoulli}(p_{row})$ and $\mathbf{u}_{col} \sim \text{Bernoulli}(p_{col})$, where $p_{row}$ and $p_{col}$ are determined based on the ratio of $|I^l|$ over the row or column size of $\mathbf{W}$.

Therefore, our approach to importance measures leverages these criteria to extract the capacity needed to finetune the pretrained model on a downstream task using LoRAs. This is the proxy for the sparsity ratio mentioned in Theorem 4.1 that allows us to sample masks while the output of both masked and unmasked LoRAs remains bounded. We refer to this approach as Partial-LoRA in the remainder of this paper. Alternatively, specific locations extracted using this method can also be used to generate masks. We name this approach Targeted-LoRA and compare against it in the experiments section.

## 5 EXPERIMENTS

**Datasets and Models.** We use OpenAI CLIP for all our image classification experiments, adding a linear classifier on top of the image features and initializing it with language prompts from the text encoder. We use OxfordIIITPet (Pets) (Parkhi et al., 2012), CIFAR-10 (Krizhevsky, 2009), CIFAR-100 (Krizhevsky, 2009), Flowers (Tung, 2020), FER2013 (Zahara et al., 2020), GTSRB (Stallkamp et al., 2011), FGVC-Aircraft (Maji et al., 2013) and ImageNet-1k (Russakovsky et al., 2015) datasets, with respective class counts of 37, 10, 100, 101, 7, 43, 102, and 1000 representing tasks with varying levels of complexity. Each dataset uses 16 shots, except Flowers, which uses 6 shots due to its smaller size. We employ a similar sized validation set for early stopping. For our tests on language models, we use Deberta-V3-Base (He et al., 2021). We perform similar experiments to vision on datasets from the GLUE benchmark (Wang et al., 2018). For the number of shots, 2% of the smaller datasets and 5% of the larger datasets are used for determining the sparsity ratios. For training on vision datasets, the same few-shot dataset for sparsity ratio derivation is used. For language, while a few-shot dataset is used for deriving the ratios, the whole dataset is used during training to match the workflow used by the SoTA methods. Models are trained with AdamW optimizer and a cosine annealing learning rate scheduler. We run all our experiments on 4 Nvidia A100 GPUs. We conduct training sessions with different seeds and initial learning rates range of {0.0001-0.005}, resulting in 20 models per method, and report mean accuracy of the top 5 models based on validation

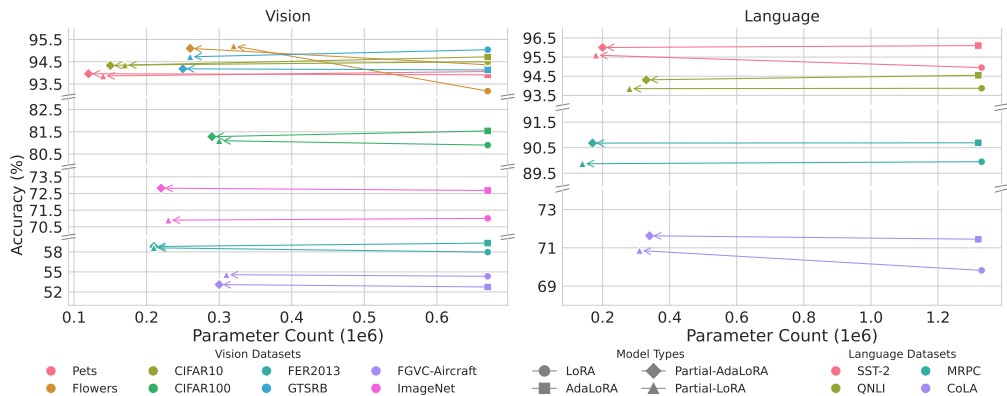

Figure 2: Accuracy against trainable parameters for LoRA, AdaLoRA, Partial-LoRA and Partial-AdaLoRA across vision and language datasets.

scores. For the language models, it should be noted that we report results on the development sets of the GLUE benchmark.

**Setting and Baselines.** In our experiments, we apply our method to both LoRA (Hu et al., 2022) and AdaLoRA (Zhang et al., 2023c) resulting in Partial-LoRA and Partial-AdaLoRA. AdaLoRA, a state-of-the-art (SotA) PEFT approach built on LoRA, adaptively adjusts the rank of low-rank residuals during training to match the task's requirements. We apply our method to both approaches as a baseline, demonstrating the potential for sparsification in any subsequent methods based on LoRA. To compare against other SotA methods focused on PEFT performance, we report results for LoRA+ and DoRA. Additionally, we include results for VeRA, another SotA PEFT method based on LoRA that focuses on reducing the number of trainable parameters. For baselines involving random masking-based pruning, we use the Pyramidal (Liu et al., 2022) and Balanced (Frankle & Carbin, 2019) random pruning methods. The Pyramidal method determines the sparsity factor for each layer based on its depth, using the formula $p_l = p^l$, where $p$ is the predetermined sparsity factor for the first layer, and $l$ represents the layer depth. The Balanced approach assigns a fixed sparsity factor to each layer, regardless of depth. Although neither method offers a principled approach for deriving the sparsity factor, and both prune the model without considering each layer's specific requirements, we still find it beneficial to visualize the impact of varying sparsity factors on accuracy and compare them to our approach.

## 5.1 SPARSIFYING LoRAs

Here we analyze the impact of sparsifying LoRA and AdaLoRA, resulting in Partial-LoRA and Partial-AdaLoRA, using methods detailed in section 4.2 through our proposed SVD-based approach. As shown in Figure 2, our method significantly reduces the number of trainable parameters across both vision and language tasks. For vision tasks, we observe an average reduction of over 60%, with a maximum of 80% for the Pets dataset. Language tasks show even greater reductions, averaging 82% with a maximum of 87% for MRPC. Importantly, these reductions are achieved while maintaining accuracy compared to the unmasked LoRA and AdaLoRA models across both modalities. An exception is the Flowers dataset where we see a significant increase in accuracy which we discuss later in section 5.2. As mentioned, the results in Figure 2 for vision datasets are obtained by training on a few-shot dataset. We also report the results of training on the whole dataset in Appendix B to show that the performance is maintained when trained on the whole dataset as well.

## 5.2 COMPARISON TO THE SoTA

We compare the results of sparsified LoRA and AdaLoRA to their full-parameter counterparts alongside DoRA, LoRA+, and VeRA as the state-of-the-art PEFT approaches. These results are visualized in Figure 3 for vision datasets and Figure 4 for language datasets. We provide the same results in quantitative format in Appendix I. LoRA+, AdaLoRA, and DoRA have roughly the same number of parameters and obtain similar performance across all datasets and modalities, with LoRA falling slightly behind in performance. This is expected since the three former methods are the more

recent PEFT approaches. As shown in section 5.1, Partial variants of AdaLoRA and LoRA obtain similar results to their full-parameter counterparts which means the performance obtained from the Partial variants is competitive with the SoTA methods with a much lower parameter count.

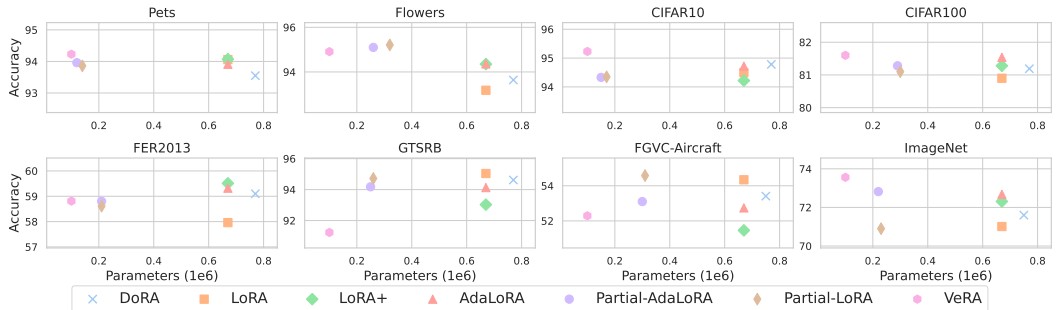

Figure 3: Comparison against the SoTA for vision datasets

Compared to VeRA, a method aimed at lowering the number of PEFT parameters, our approach obtains competitive results while significantly outperforming this method on GTSRB and FGVC-Aircraft datasets. We attribute this to how our method can strike a balance between the number of parameters and performance using the pretrained model as a source for deriving the capacity needed for finetuning. Moreover, VeRA can only modify the number of parameters through changing the rank of the low-rank matrices and sharing low-rank matrices across layers.

While both can be done using our approach, our method also allows for sparsification of the low-rank matrices themselves. Therefore, using the accuracy threshold discussed in section 4, another degree of freedom is added to allow for more flexibility in sparsification of PEFT residuals. The performance on Flowers is one of the more notable ones across all datasets since the methods with lower parameter counts obtain a better performance. We believe this is due to the overparameterization of other methods causing overfitting since Flowers has a much smaller training set compared to the other datasets. As mentioned in section 4.1, the preservation of flow from the pruning literature may be a concern for the sparsified LoRAs due to how the preserved weights are determined for each layer in an isolated manner independent of the other layers. To show this is not a concern for our work due to the additive nature of LoRA residuals, we report the results of an ablation study where we compare the accuracy when employing flow preservation techniques. These results are provided in Appendix H.

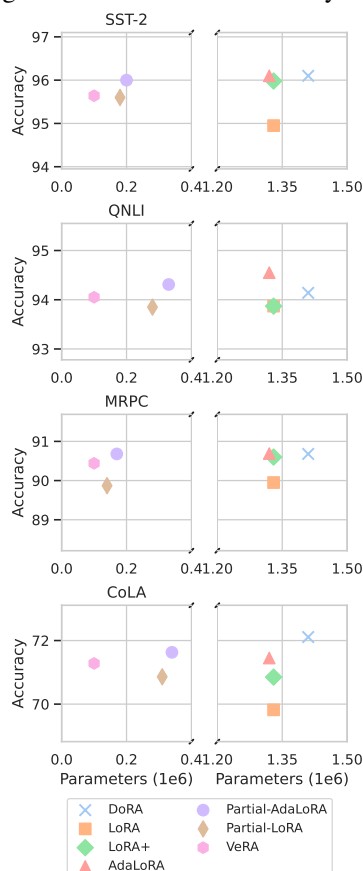

Figure 4: Comparison against the SoTA for language datasets

### 5.3 SWEEPING THROUGH SPARSITY RATIOS

We use two methods to sweep through different sparsity ratios to observe the obtained accuracy across different number of parameters. To this end, we use Balanced and Pyramidal pruning methods. For the Balanced method, we sweep the sparsity ratio on the range of 0.1 to 0.9 and use it at each turn to prune every layer with the same sparsity. For Pyramidal pruning, the sparsity is increased exponentially as depth increases to allow for more parameters at the start and fewer at the later layers. As shown in Figure 5, the Pets dataset allows for a consistent reduction in the number of parameters without a significant loss in performance. This is while Flowers shows improvements with sparsity which confirms that overparameterization was leading to overfitting as discussed in section 5.2. Note that while these visualizations do confirm that the reduction in PEFT parameters does not lower performance, the Balanced and Pyramidal methods do not necessarily provide a way to derive the sparsity ratio before sweeping.

Table 2: Different sparsity ratio derivations

| Dataset | LoRA | Targeted-LoRA (SVD) | (SVD) | Partial-LoRA (SNIP) | (IMP) | Inverted |
|---|---|---|---|---|---|---|
| Pets | 94.06 | 93.44 | 93.86 | 93.95 | 93.82 | 94.00 |
| Flowers | 93.18 | 95.01 | 95.21 | 95.24 | 94.48 | 93.71 |
| CIFAR-10 | 94.50 | 94.45 | 94.35 | 94.22 | 93.89 | 94.59 |
| CIFAR-100 | 80.90 | 80.85 | 81.10 | 80.97 | 80.51 | 80.58 |
| FER2013 | 57.96 | 58.51 | 58.61 | 58.86 | 57.94 | 58.43 |
| GTSRB | 95.04 | 94.64 | 94.72 | 94.53 | 94.90 | 94.40 |
| FGVC-Aircraft | 54.34 | 53.44 | 54.58 | 55.09 | 54.21 | 51.86 |
| **Average** | 81.42 | 81.48 | 81.78 | 81.84 | 81.39 | 81.08 |

## 5.4 LARGER MODELS

In this section, we experiment with LLAMA2-7b to investigate the effectiveness of our approach on large language models. Table 1 shows that compared to the full fine-tuning and unmasked LoRAs, our approach obtains similar accuracy and the performance remains close to the unmasked counterpart.

While (Xu & Zhang, 2024) resorts to sweeping the sparsity ratios to find the 0.001% sparsity, through our sparsity ratio derivation method proposed in Section 4.2.1, we obtain a sparsity of

Table 1: Evaluation Using LLAMA2-7b

| Task | FT | LoRA | Masking (0.001%) | Partial-LoRA (0.001%) |
|---|---|---|---|---|
| SST-2 | 94.7 (6.7B) | 95.4 (4.2M) | 95.5 (68K) | 94.9 (68K) |

0.001% without the need for searching over this parameter or training the mask itself. Meanwhile, our method with the derived sparsity achieves a comparable performance to (Xu & Zhang, 2024), as the SotA masking approach.

## 5.5 SPARSITY RATIO DERIVATION ABLATION

**Partial-LoRA and Targeted-LoRA:** As mentioned in section 4.2.2, while the results in section 5.1 are obtained using random sampling of masks, the specific locations of the weights derived by the importance measuring criteria $I^l$ may also be used to generate masks which we call Targeted-LoRA. Regardless of the dataset, the number of parameters for Targeted-LoRAs is the exact same as Partial-LoRAs. We report the results of Targeted-LoRA in Table 2.

In terms of average accuracy, Targeted-LoRA scores similar to LoRA while slightly falling behind Partial-LoRA. The case-by-case accuracy of both approaches is also close for most datasets. However, for FGVC-Aircraft, Partial-LoRA outperforms Targeted-LoRA by 1%, with Targeted-LoRA falling behind LoRA by a similar margin. This shows that targeting specific subnetworks is unnecessary and the capacity required at each layer for the downstream task is the only important factor. We provide a deeper analysis of the behavior of Partial-LoRA and Targeted-LoRA compared to LoRAs in Appendix C. We use the norm of the residuals to show the similarity of Targeted and Partial methods in the magnitude of residuals across layers. We show that the changes in magnitude for both these methods across layers, while being similar to each other, vary compared to LoRAs signaling potential differences in the finetuning dynamic. Additionally, to explore a middle ground between random and deterministic sampling of masks, we treat the importance values as a distribution and sample subnetworks from this distribution. The result of training these subnetworks is provided in Appendix D where we show slight improvements in accuracy compared to Partial and Targeted-LoRAs.

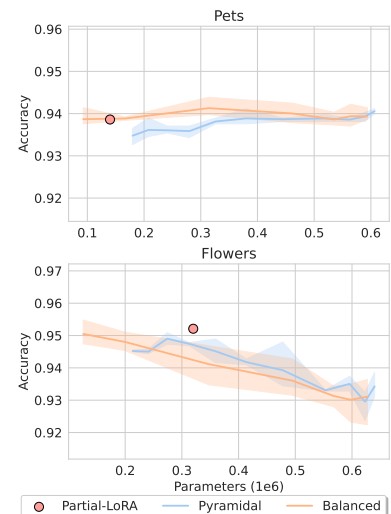

Figure 5: Sweeping sparsity ratio.

**Importance Measuring Criterion:** Aside from the proposed method in section 4.2.2, we can also use aforementioned importance measuring criterion from the pruning literature such as SNIP or IMP to infer the sparsity ratio. The columns named SNIP and IMP under Partial-LoRA in Table 2 represent the results of these methods. SVD and SNIP-based methods obtain a similar performance while IMP

falls slightly behind due to a lower performance across multiple datasets. To get a deeper insight into the similarity between these methods, we visualize the overlap of the subnetworks extracted by these methods in a pair-wise setting. This visualization is provided in Appendix E where there is a large overlap between the three methods in the first 6 layers of the model. The relatively smaller overlap in the later layers is mainly due to how small the size of the subnetwork itself is. We provide a visualization on the relative size of the subnetworks across layers for all three importance measures in Appendix G. The initial layers have the smallest sparsity ratio for most datasets while this sparsity grows as we move towards the final layers. For Pets, the sparsity is similar across all layers potentially due to the high performance of the pretrained model prior to finetuning.

**Inverted Masks:** The proposed sparsity ratio derivation methods in the case of Targeted-LoRA generate masks that result in sparse residuals for the adapter. As an ablation, we run the same experiments with these masks inverted. This way, a significant reduction in the fine-tuned performance would show the importance of the masks. Due to how sparse Targeted-LoRAs are, the inverted masks will result in a larger number of parameters during training compared to Partial-LoRAs. The results are reported in Table 2. In the case of Pets and CIFAR-10, the performance of the inverted masks is better than that of Targeted-LoRA while for the rest this method results in competitive or degraded performance. Overall, this shows targeting specific subnetworks of the pretrained model does not necessarily result in better performance.

## 5.6 ABLATION ON THE LORA RANK

We examine whether the results from section 5.1 depend on the LoRA rank. We set LoRA ranks to 1, 4, 16, 32, and 64, showing the average performance as a solid line alongside the maximum and minimum performance based on multiple training sessions in Figure 6 for CIFAR-100 and FGVC-Aircraft, with CIFAR-10 and GTSRB in Appendix F. Although the top-performing method varies by dataset, the relative behavior observed in section 5.1 remains consistent. Partial and Targeted methods do not significantly degrade performance and improve it for FGVC-Aircraft across most ranks. Thus, the results are not dependent on the rank.

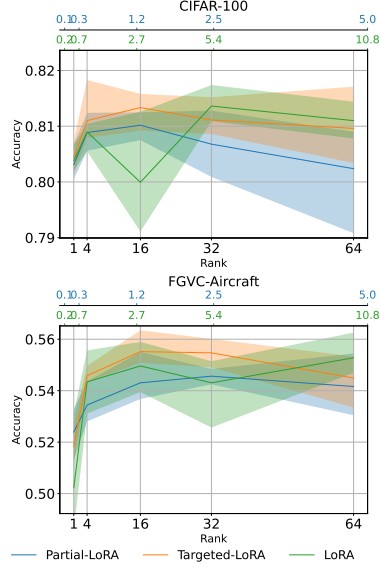

Figure 6: Rank ablation with parameter count for each dataset in millions.

## 6 DISCUSSION AND CONCLUSION

In this study, we explored the efficacy of random masking in LoRAs for fine-tuning large pre-trained models. Building on the lottery ticket hypothesis, we showed that highly sparse "winning ticket" subnetworks within LoRA and its state-of-the-art variants can match the performance of fully parameterized counterparts. Our experiments on vision and language modalities confirmed that our proposed approach named Partial-LoRAs can achieve similar performance to SotA methods with significantly fewer trainable parameters, reducing them by up to 87%. Key contributions include establishing a theoretical basis for lottery tickets within LoRA residuals and demonstrating the effectiveness of random masking. We found that targeting specific locations of large pretrained models for parameter-efficient finetuning in an effort to prune LoRAs is unnecessary. As long as the per-layer sparsity of the LoRA residuals is kept within a bound determined by our proposed sparsity ratio derivation method, the performance of pruned LoRAs is maintained while the number of parameters decrease significantly. Our ablation studies further validated the robustness of random masking strategies. We showed that strategies like preserving activation flow between layers, crucial for full model pruning, are unnecessary for LoRA masking due to the residual update formulation. Our method can be implemented on top of existing and future LoRA-based PEFT methods by pruning the components of the low-rank matrices as shown in this work to maintain the advantages of those works while reducing the computational load. Our work highlights the potential for significant computational savings in parameter-efficient fine-tuning. We pave the way for more sustainable and accessible deployment of large pre-trained models. Future work will optimize sparsity factors, exploring the middle ground between deterministic and stochastic pruning methods. Another potential future trajectory is the analysis of the training dynamics for Partial-LoRAs.

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

## A    PROOF OF THEOREM 1

**Theorem A.1** *Define a network $f_T$ of depth $L$ parameterized by pretrained weights and biases $W_l$ and $b_l$ with low-rank adapters at each layer parameterized by the residuals $\Delta W_l^T$. Additionally, define a pruned network $f_{LoRA}$ of depth $L + 1$ parameterized by $\Delta W_l \cdot U$ and the same $W_l$, $b_l$ as the target model. The pruned model consists of sparsity factors $p_l$ for layer $l$ and $\Delta w_{ij}^l \sim U([-1, 1])$. Both networks have $n_{T,l}$ and $n_{LoRA,l}$ neurons at layer $l$. Then given variables $\epsilon, \delta \in (0, 1)$, with failure probability $1 - \delta$, it holds that $\max_{x \in \mathcal{D}} \|f_T(x; \Delta W^T) - f_{LoRA}(x; \Delta W \cdot U)\| \le \epsilon$ for compact space $D$, if*

$$n_{LoRA,l} \ge C \frac{n_{T,l}}{\log(1/1 - p_{l+1})} \log\left(\frac{1}{\min(\epsilon_l, \delta/\rho)}\right) \tag{7}$$

**Proof**   We extend the proof by (Gadhikar et al., 2023) for the existence of strong lottery tickets (SLTs) in Erdős-Rényi (ER) networks to low-rank adapters. As mentioned in Section 3, for each layer of a model, after the application of the low-rank adapter the output of the layer previously defined as $f(x) = Wx + b$ becomes $f(x) = (W + \Delta W)x + b$ where $W$ and $b$ are the weight and bias of the pretrained layer and $\Delta W$ represents the learnable residuals. Therefore for both the pruned LoRA and target LoRA we have the following.

$$f_T(x) = (W + \Delta W^T)x + b. \tag{8}$$

Whereas for the pruned layer, we have,

$$f_{LoRA}(x) = (W + \Delta W \cdot U)x + b. \tag{9}$$

Where $U$ represents the mask randomly sampled from a Bernoulli distribution. Given that $W$ and $b$ are the same frozen parameters from the pretrained model, and hence will remain unchanged throughout the training and across both LoRA and PaLoRA, we have the following.

$$f_T(x) - f_{LoRA}(x) = \Delta W^T x - (\Delta W \cdot U)x \tag{10}$$

The remaining elements of $f_T$ and $f_{LoRA}$ can be rephrased as new layers without a bias parameter with the weights of the residuals and $Wx + b$ as the new bias parameter. Therefore according to (Gadhikar et al., 2023),

$$\mathbb{P}[\max_{x \in \boldsymbol{D}} \|g_T(x; \Delta W^T) - g_{LoRA}(x; \Delta W \cdot U)\| \le \epsilon] \ge 1 - \delta. \tag{11}$$

Where $\epsilon, \delta \in (0, 1)$, $g_T$ and $g_{LoRA}$ represent the new layers, $\rho = \frac{CN_T^{1+\gamma}}{\log(1/(1-\min_l p_l))^{1+\gamma}} \log(1/\min\{\min_l \epsilon_l, \delta\})$ for any $\gamma \ge 0$ with $C$ as a distribution dependent constant. $N_T$ represents the number of non-zero parameters of the model. $\epsilon_l$ is defined as follows (Burkholz, 2022).

$$\epsilon_l = \frac{\epsilon}{n_{LoRA,L}L}\left[(1 + B_{l-1})\left(1 + \frac{\epsilon}{L}\right)\prod_{k=l+1}^{L-1}\left(\left\|W_{(k)}^T\right\|_\infty + \frac{\epsilon}{L}\right)\right]^{-1}, B_l := \sup_{x \in \mathcal{D}}\left\|\boldsymbol{x}_{LoRA}^{(l)}\right\|_1. \tag{12}$$

Where $x_{LoRA}^l$ represents the features of the pruned LoRA at layer $l$.

## B    FROM LOW DATA REGIME TO TRAINING ON THE WHOLE DATASET

As shown in section 5.1, the number of trainable parameters is significantly reduced for for Partial-LoRAs. Here, we assess if the performance similarity between LoRAs and Partial-LoRAs is due to the large number of parameters relative to the small training dataset size. Table 3 shows the performance of each method compared to LoRAs when trained on the entire training set $\mathcal{D}$ instead of a few-shot subset $\mathcal{D}_t$. The performance similarity between the three methods in Table 3 is consistent with the few-shot results in section 5.1. Therefore, pruned LoRAs perform similarly to LoRAs, regardless of dataset size.

Table 3: Performance of training on the whole dataset

| Dataset | LoRA | Targeted-LoRA | Partial-LoRA |
|---|---|---|---|
| CIFAR-10 | 96.68 | 97.68 | 97.79 |
| CIFAR-100 | 87.16 | 87.47 | 87.59 |
| GRSRB | 99.12 | 99.02 | 99.08 |
| FGVC-Aircraft | 65.61 | 64.93 | 65.65 |
| Average | 87.14 | 87.28 | 87.53 |

## C    MAGNITUDE OF RESIDUALS

While LoRAs modify all weights, Partial-LoRAs modify only a small subset. We study the impact of this reduction in the number of trainable parameters on the magnitude of the residuals obtained through fine-tuning. Figure 7 shows the norm of the residual matrix across the 12 layers of the transformer model, focusing on the first fully connected layer of each block, with additional visualizations for CIFAR-100 and GTSRB datasets visualized in Figure 8. Moreover, Figure 9 and Figure 10 visualize the same norms for the second fully connected layer and attention projection layers, respectively. Alongside Partial-LoRA and Targeted-LoRA derived using our proposed SVD-based method, we also provide the visualization for SNIP and IMP based Targeted-LoRAs as well.

The norm of the residual matrix from Targeted-LoRA behaves similarly to the Partial-LoRA method across all importance measures, indicating the importance of the number of modified parameters over the specific elements. Additionally, we note differences between LoRAs and masked LoRAs. Neither masking approach achieves magnitudes similar to LoRAs, changing the model's overall behavior. For example, LoRA shows an uptick in norm for the final layer of the Pets dataset, while masked models show the opposite. In CIFAR-10, LoRA maintains consistent norm values across layers, whereas masked LoRAs exhibit drastic changes. This pattern remains consistent across the datasets with the norm value going down significantly for the final layer. Additionally, the norm for Partial-LoRA follows the Targeted method closely across each layer for every importance measure. This gives us insights on how the randomly masked LoRAs bearing the same capacity as the Partial LoRAs can modify the pretrained weights in a similar manner.

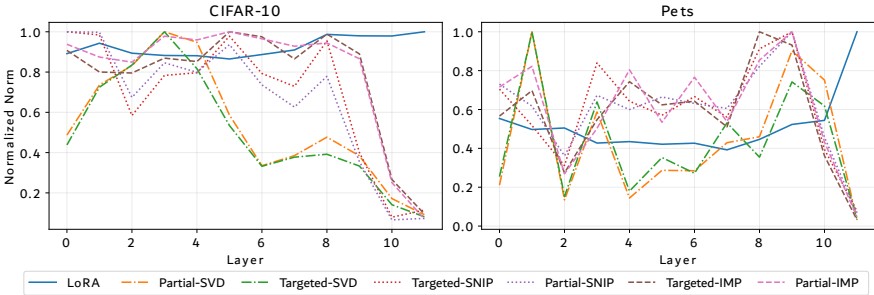

Figure 7: Normalized magnitude of residualsfor CIFAR-10 and Pets for the first fully-connected layer of every transformer block.

Here we provide the visualization of the residual norms for two other datasets that are CIFAR-10 and GTSRB.

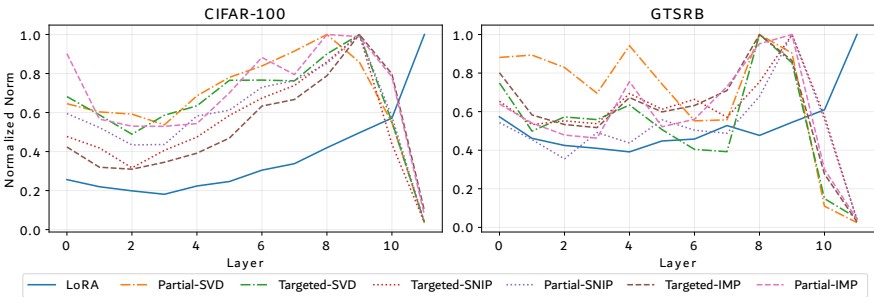

Figure 8: Magnitude of the residuals for datasets CIFAR-100 and GTSRB for the first fully-connected layer of every transformer block. The magnitude goes down as layer depth increases.

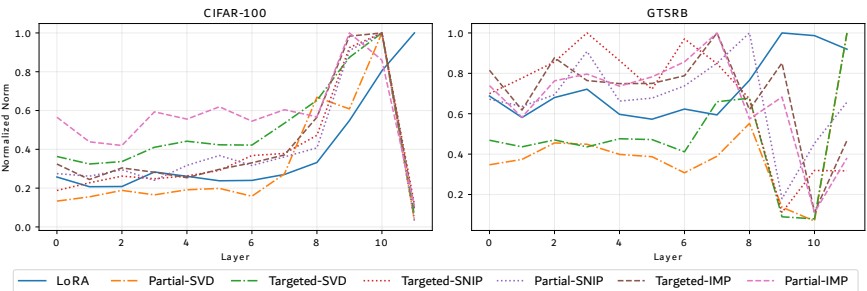

Figure 9: Magnitude of the residuals for datasets CIFAR-100 and GTSRB for the second fully-connected layer of every transformer block. The general pattern follows that of the first layer of the transformer block.

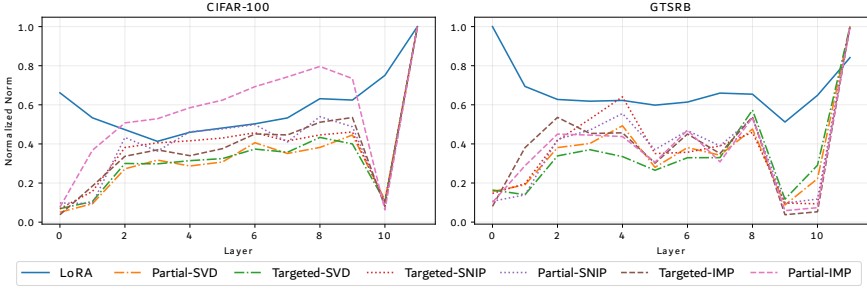

Figure 10: Magnitude of the residuals for datasets CIFAR-100 and GTSRB for the attention layer (inward projection) of every transformer block. The pattern is opposite of that of the fully-connected layers. This could be due to how the attention layer is itself made of 3 layers, namely, the query, key and value projections.

# D EXPLORING DETERMINISTIC AND STOCHASTIC UTILIZATION OF IMPORTANCE MEASURES

In Section 5.1, Targeted-LoRA methods used deterministic approaches to select elements for modification based on importance scores. Here, we treat these scores probabilistically, sampling indices for modification. We introduce randomness by passing importance values through a softmax function and explore the effects by varying the temperature parameter. Then, we sample masks using the derived distribution. Table 11 shows that the stochastic IMP approach achieves accuracy close to LoRAs

for Pets and improves performance on CIFAR-100 across temperatures. However, the SVD-based method's performance degrades with randomness, possibly due to its dependence on the SVD of the pretrained matrix, unlike the IMP method, which extracts scores from gradient magnitudes, allowing for a more meaningful interpretation of this stochasticity.

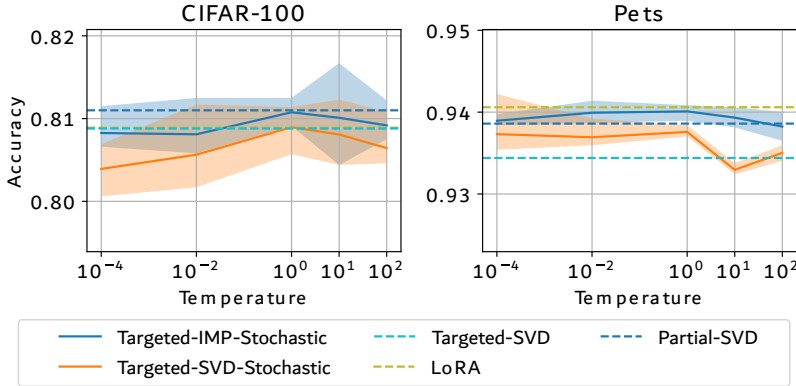

Figure 11: Stochastic approach to subnetwork extraction has minimal effects on performance.

## E    AGREEMENT IN THE SUB-NETWORK EXTRACTION.

In this section, we aim to determine the similarity between sub-networks extracted by SVD, SNIP, and IMP. We hypothesize that if there is a meaningful set of elements in each weight matrix responsible for a model's performance, there would be some overlap between the sub-networks identified by these different methods. The results for CIFAR-10, Pets, CIFAR-100 and GTSRB are provided in the following.

**Overlap of layers on different methods:** In Figure 12, the overlap for CIFAR-10 and Pets is larger than 50% in the first 6 layers of the model when we compare the sub-network extracted by SVD-based and gradient-based methods. This suggests that using importance measures to infer the number of elements to modify is effective, highlighting the similarity between the SVD-based and gradient-based methods. Similar results are shown in Figure 14 for CIFAR-100 and GTSRB. Interestingly, the overlap between the SVD-based method and the gradient-based methods is different from the other datasets. For Pets, the overlap is consistently low across the layers. This could potentially be due to the fine granular information required to be modified by the LoRA. If so, a larger portion of the top singular vectors would be used for the subnetwork extraction phase, leading to a smaller overlap across all layers.

**Overlap of layers on the same method but different shots:** We compare each method using different shots from each dataset to complete these observations. Shared concepts should be reflected in overlaps of sub-networks across different shots. As shown in Figure 13, the SVD-based method shows a large overlap between different seeds, with full overlap for GTSRB across all layers. This is expected since SVD starts with the top singular vectors. Surprisingly, the gradient-based method also shows significant overlap across different shots for both datasets. Therefore, considering the results in Table 2, while specific indices may not be optimal for training, the sparsity factor can be effectively determined using importance measuring methods, as shown by the comparisons between different importance methods. Similar results are shown in Figure 15 for CIFAR-100 and GTSRB.

## F    RANK ABLATION STUDY FOR OTHER DATASETS

In this section, we provide the visualizations for the ablation study on the rank of the low-rank residuals for CIFAR-10 and GTSRB. Fig. 16, visualizes the accuracy values across ranks of 1, 4, 16, 32 and 64. Similar to the results for CIFAR-100 and FGVC-Aircraft, the relative performance

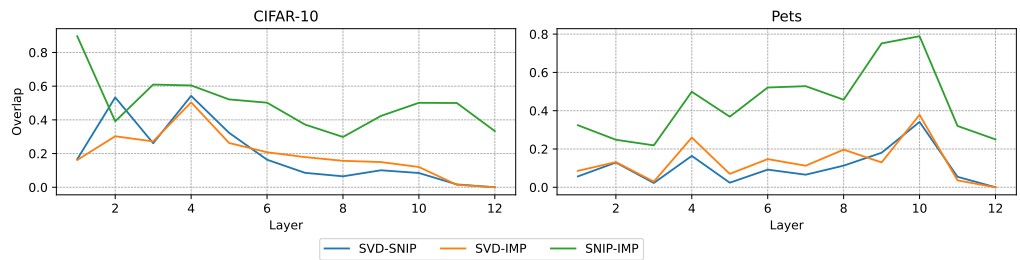

Figure 12: Overlap of top indices across different importance measures for CIFAR-10 and Pets.

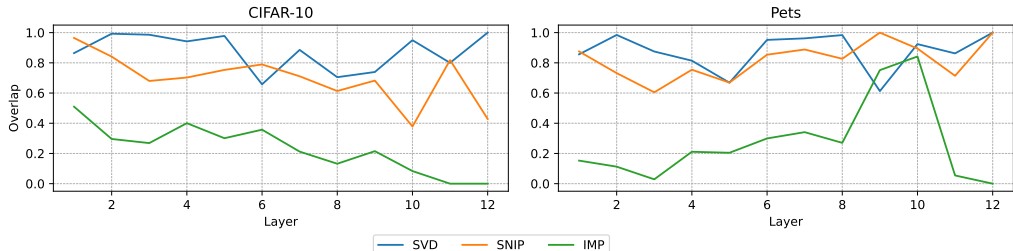

Figure 13: Overlap of top indices across different shots for CIFAR-10 and Pets.

between LoRAs and pruned counterparts is consistent showing that performance of pruned LoRAs is not dependent on rank.

## G  IMPORTANT ELEMENTS ACROSS DIFFERENT LAYERS

In this section, we visualize the fraction of elements in the subnetwork extracted by SNIP, SVD and IMP across the layers. These visualizations are provided in Figure 17. As mentioned in Appendix E, the number of elements chosen to be preserved in the extracted subnetwork decreases as depth increases, causing a smaller overlap between the importance measures.

## H  EFFECTS OF FLOW PRESERVATION

In the pruning literature, it is common to modify the masks resulting from importance measures to preserve the flow of information throughout the layers of the model. This is due to the possibility that if two layers are pruned independently of each other, the activations resulting from one layer might encounter weights that have been masked out, leading to a vector of zeros as the output of the second layer. Here we show that in the fine-tuning setting where LoRAs adjust the pretrained layers, this step is not necessary. We use the IMP importance method and infer the masked elements for each layer independent of other layers. Then, we use the same approach but for each layer, the calculated scores are added to the scores of the next layer to reweigh the importance of each element in favor of important elements from the next layer. The performance of both methods is reported in Table 4. The results with flow preservation are named Continuous and the results without this alteration of the importance measure are named Isolated. Both the individual datasets and the overall average are consistently close confirming that flow preservation is not necessary in the case of our work.

## I  DETAILED RESULTS FROM QUANTITATIVE EVALUATION

Here we provide the same results from Figure 3 and Figure 4 in tabular format. Table 5 and Table 6 show the performance of each method on each dataset tested in our work.

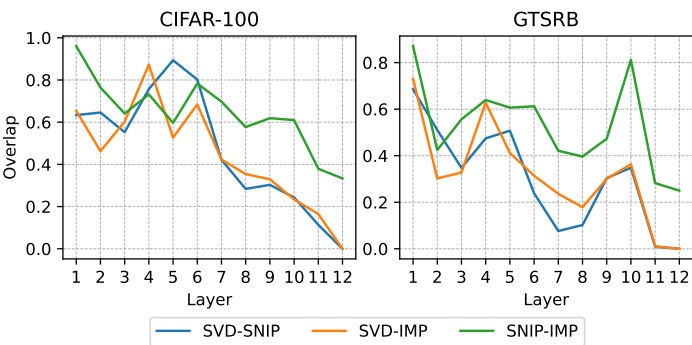

Figure 14: Overlap of top indices across different importance measures for CIFAR-100 and GTSRB.

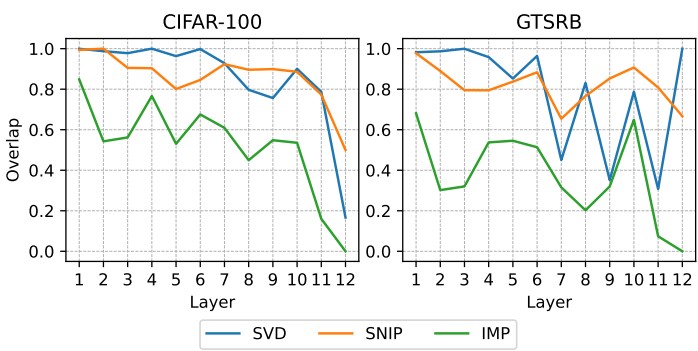

Figure 15: Overlap of top indices across different shots for CIFAR-100 and GTSRB.

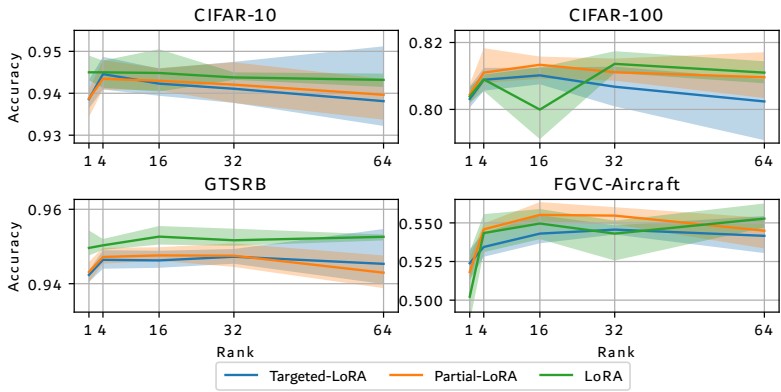

Figure 16: Ablation on the rank of the low-rank residual weights. Regardless of rank, random masking allows for reduction in the number of parameters without significant reduction in performance.

Table 4: Comparison of Isolated and Continuous Data Accuracy

| Dataset | Isolated (%) | Continuous (%) |
| --- | --- | --- |
| Pets | 93.82 | 93.64 |
| CIFAR-10 | 93.89 | 94.20 |
| CIFAR-100 | 80.51 | 80.86 |
| GTSRB | 94.90 | 94.75 |
| FGVC-Aircraft | 54.21 | 54.67 |
| Average | 83.47 | 83.58 |

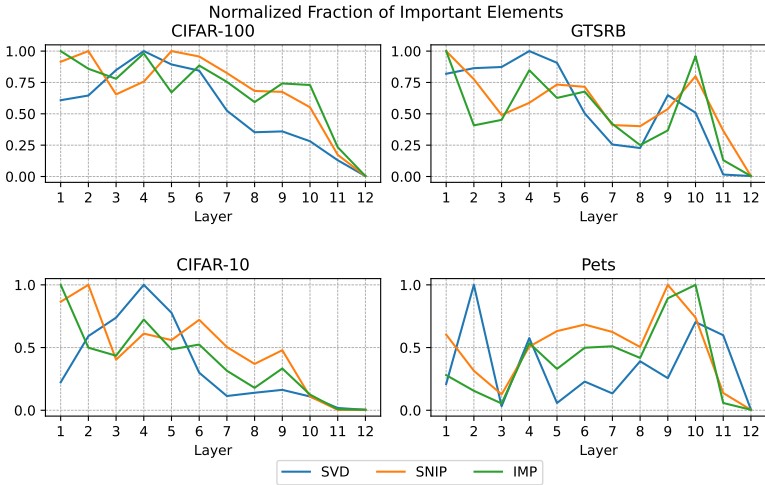

Figure 17: Fraction of the number of elements in the extracted subnetwork from different importance measures over the number of parameters in the pretrained model. The number of important elements goes down in deeper layers suggesting narrowing of the high-level knowledge required to infer accurately on a task.

Table 5: Performance Comparison Across Vision Tasks (Parameter Count)

| Method | Pets | Flowers | CIFAR10 | CIFAR100 | FER2013 | GTSRB | FGVC-Aircraft | ImageNet |
|---|---|---|---|---|---|---|---|---|
| LoRA | 94.06 (0.67M) | 93.18 (0.67M) | 94.50 (0.67M) | 80.90 (0.67M) | 57.96 (0.67M) | 95.04 (0.67M) | 54.34 (0.67M) | 71.01 (0.67M) |
| Partial-LoRA | 93.86 (0.14M) | 95.21 (0.32M) | 94.35 (0.17M) | 81.10 (0.30M) | 58.61 (0.21M) | 94.72 (0.26M) | 54.58 (0.31M) | 70.90 (0.23M) |
| VeRA | 94.23 (0.10M) | 94.91 (0.10M) | 95.23 (0.10M) | 81.60 (0.10M) | 58.81 (0.10M) | 91.23 (0.10M) | 52.29 (0.10M) | 73.56 (0.10M) |
| LoRA+ | 94.07 (0.67M) | 94.35 (0.67M) | 94.22 (0.67M) | 81.28 (0.67M) | 59.51 (0.67M) | 93.03 (0.67M) | 51.46 (0.67M) | 72.31 (0.67M) |
| DoRA | 93.55 (0.77M) | 93.64 (0.77M) | 94.78 (0.77M) | 81.19 (0.77M) | 59.10 (0.77M) | 94.62 (0.77M) | 53.41 (0.75M) | 71.60 (0.75M) |
| AdaLoRA | 93.91 (0.67M) | 94.36 (0.67M) | 94.72 (0.67M) | 81.54 (0.67M) | 59.32 (0.67M) | 94.13 (0.67M) | 52.74 (0.67M) | 72.68 (0.67M) |
| Partial-AdaLoRA | 93.38 (0.12M) | 94.62 (0.26M) | 94.11 (0.15M) | 81.02 (0.29M) | 58.77 (0.21M) | 93.79 (0.25M) | 52.22 (0.30M) | 72.79 (0.22M) |

Table 6: Performance Comparison Across Language Tasks (Parameter Count)

| Method | SST-2 | QNLI | MRPC | CoLA |
|---|---|---|---|---|
| LoRA | 94.95 (1.33M) | 93.87 (1.33M) | 89.95 (1.33M) | 69.82 (1.33M) |
| Partial-LoRA | 95.60 (0.18M) | 93.85 (0.28M) | 89.87 (0.14M) | 70.86 (0.31M) |
| Partial-AdaLoRA | 96.00 (0.20M) | 94.31 (0.33M) | 90.68 (0.17M) | 71.63 (0.34M) |
| Vera | 95.64 (0.10M) | 94.05 (0.10M) | 90.44 (0.10M) | 71.28 (0.10M) |
| LoRA+ | 95.98 (1.33M) | 93.87 (1.33M) | 90.60 (1.33M) | 70.85 (1.33M) |
| DoRA | 96.10 (1.41M) | 94.14 (1.41M) | 90.68 (1.41M) | 72.11 (1.41M) |
| AdaLoRA | 96.10 (1.32M) | 94.55 (1.32M) | 90.69 (1.32M) | 71.45 (1.32M) |

