# OpenReview forum: "The Quest for Winning Tickets in Low-Rank Adapters"
_ICLR.cc/2025/Conference — Submitted to ICLR 2025_

### Official Review · Reviewer_vQUE · 2024-10-28

**Soundness:** 3
**Presentation:** 2
**Contribution:** 2
**Rating:** 5
**Confidence:** 4

**Summary:**

This paper explores applying the Lottery Ticket Hypothesis (LTH) to Low-Rank Adaptation (LoRA) and proposes that LoRA’s low-rank adapters contain sparse, essential subnetworks, or 'winning tickets. The authors introduce Partial-LoRA, which identifies sparse subnetworks in LoRA matrices to reduce trainable parameters while retaining model performance. Through a series of experiments on vision and language tasks, they demonstrate that Partial-LoRA can achieve notable parameter reductions with minimal loss in accuracy. The paper presents a practical approach for enhancing parameter efficiency in model fine-tuning.

**Strengths:**

1. By leveraging Partial-LoRA, the work demonstrates that it’s possible to reduce trainable parameters by up to 87% while maintaining accuracy. This reduction supports resource-efficient fine-tuning, which is highly beneficial for deploying large models on devices with limited computational capacity.

2. The authors conduct comprehensive experiments across vision and language tasks, providing quantitative results that support the claimed parameter reduction benefits. This thorough testing on multiple datasets strengthens the paper’s claims regarding Partial-LoRA's performance consistency.

3. The work provides theoretical basis for applying LTH to LoRA, detailing how randomly masked adapters can achieve performance close to fully parameterized adapters. This theory analysis contribution enhances the credibility and relevance.

**Weaknesses:**

(1) **Absence of Adaptive Masking Mechanism.** Unlike recent techniques that employ adaptive sparsity adjustments per layer [1], this paper applies a fixed, random masking ratio for identifying "winning tickets." This lack of adaptivity may lead to suboptimal performance in tasks where layer importance varies significantly.
﻿

(2) **Inadequate Theoretical Justification.** Although the authors discuss some theoretical underpinnings, there’s no formal analysis or proof of generalization guarantees for Partial-LoRA. Recent works typically provide theoretical bounds or assumptions [2], which help predict how sparse adapters might perform across different tasks or model architectures.
﻿

(3) **Lack of Robustness and Noise Testing.** The paper does not address Partial-LoRA's robustness under noisy or adversarial conditions, which is often included in parameter-efficient adaptation research. Testing under noise or adversarial setups would demonstrate the approach's practical resilience.


(4) **Limited Reproducibility Information.** The paper lacks essential implementation details, such as hyperparameter settings, initialization strategies, and optimization schedules, which hinders reproducibility. Additionally, a provided open-sourced code repository will be beneficial to enhance the reproducibility.
﻿

(5) **No Exploration of Cross-Domain Transferability.** While Partial-LoRA performs well in selected domains, there’s no assessment of how it handles cross-domain adaptation (e.g., high-resource to low-resource tasks), a feature increasingly addressed in recent LoRA adaptations. This gap limits the understanding of the method’s robustness in varied application scenarios.
﻿

**References.**

[1] Panda, Ashwinee, et al. "Lottery ticket adaptation: Mitigating destructive interference in LLMs." arXiv preprint arXiv:2406.16797 (2024).


[2] Zeng, Yuchen, and Kangwook Lee. "The Expressive Power of Low-Rank Adaptation." in the Twelfth International Conference on Learning Representations (ICRL), 2024.

**Questions:**

**1. Rationale for Random Masking Selection.** Could you clarify the choice of random masking as opposed to an adaptive sparsity mechanism, which recent studies have shown to be beneficial for task-specific and layer-specific adaptation? How does this fixed, non-adaptive approach account for variations in parameter importance across layers or tasks, and was any analysis conducted to validate the effectiveness of random masks for achieving optimal performance?


**2. Sparsity Ratio Derivation and Consistency.** The paper mentions deriving sparsity ratios based on a subset of labeled data. Could you provide details on how this subset was chosen and whether its selection impacts the stability of the resulting sparsity patterns? Additionally, is there consistency in performance if the sparsity ratios are derived from different subsets, or does it introduce variability in results?


**3. Layer-Wise Performance Variability.** Did you observe any significant performance differences across layers when applying a uniform sparsity ratio? Recent studies suggest that certain layers contribute more critically than others in large models. How did you assess the impact of sparsity on each layer, and were any experiments conducted to explore layer-specific adaptations or sparsity adjustments?


**4. Missing comparison with existing Lottery Ticket-based LoRA methods.** The authors should include existing Lottery Ticket Hypothesis-based LoRA approaches [a] [b] [c] in the related works section and provide a comparative discussion in the experimental results section. Specifically, it would be beneficial to discuss how the proposed Partial-LoRA differs from and compares to these established techniques, highlighting unique contributions and performance differences.


**5. Impact on Model Transferability Across Domains.** While Partial-LoRA performs well within specific task domains, there is limited evidence on its cross-domain performance, such as transfer from high-resource to low-resource settings. Were any experiments conducted to assess its effectiveness in such transfer scenarios? This would help clarify whether Partial-LoRA’s masking strategy generalizes well across diverse data distributions.

**6. Absence of Robustness Validation.** Given the potential application of Partial-LoRA in real-world tasks, was any consideration given to its robustness under adversarial attacks or noisy data? How does the model handle sparsity under these conditions? Many recent methods include robustness testing to validate performance stability, and it would be helpful to understand how Partial-LoRA performs in this regard.


**7. Implementation Details on Hyperparameters.** There is minimal detail on hyperparameter selection, initialization settings, and optimization schedules. Also, it is suggested the authors provide implementation codes of your proposed Partial-LoRA.

**References.**

[a] Panda, Ashwinee, et al. "Lottery ticket adaptation: Mitigating destructive interference in LLMs." arXiv preprint arXiv:2406.16797 (2024).

[b] Xu, Jing, and Jingzhao Zhang. "Random Masking Finds Winning Tickets for Parameter Efficient Fine-tuning." arXiv preprint arXiv:2405.02596 (2024).

[c]  Yuan, Fei, et al. "KS-Lottery: Finding Certified Lottery Tickets for Multilingual Language Models." arXiv preprint arXiv:2402.02801 (2024).

---

> ### Author Response · Authors · 2024-11-25
>
> Thank you for your thoughtful evaluation and feedback. We appreciate your recognition of the novelty and potential utility of Partial-LoRA, as well as your detailed suggestions for improving the scope and rigor of our work.
>
> Due to space constraints, we'll address the weaknesses section here:
>  - **W1 - Adaptive Masking Mechanism**: Our method employs Algorithm 1 in the paper **to every layer separately**. For attention layers, we apply our method to query, key, and value weights separately as well. Therefore, the sparsity ratio is already adaptive to the layer at hand and is not set for the whole model or for multiple layers at once. This means we do not require adaptive mechanism since **our method is naturally adaptive**.
>  - **W2 - Theoretical Justification**: Our theoretical results backed by the empirical analysis on both vision and language tasks show that partial-LoRAs can approximate LoRAs. Thereby, our work is focused on the approximation of LoRAs and masking low-rank adapters. Investigating generalization guarantees would be out of the scope of this paper.
>  - **W3, Q6 - Robustness and Noise Testing**: We consider testing under adversarial conditions or noisy data to be out of the scope of this work. As mentioned, our work is concerned with approximation of LoRAs and showing that pruning LoRAs can allow for significant parameter reductions without performance drops.
>  - **W4, Q7 - Reproducibility Information**: We provide implementation details and the testing scenarios in Section 5. Additionally, as mentioned, our approach is based on pruning adapters such as LoRA and AdaLoRA. For each, we use the same settings used in the respective works for a fair comparison. We will make this more clear in the text.
>  - **W5 - Cross-Domain Transferability**: We're not sure what is exactly being asked here. Additionally, no works are cited for this question. We provide results in different modalities on datasets of different granularity with different sizes.

---

> > ### Author Response · Authors · 2024-11-25
> >
> > We address the questions section here:
> >
> >  - **Q1 - Random Masking Selection**: Our method is naturally adaptive due to the changes in the sparity ratio per layer. Additionaly, we proivide test named Targeted-LoRA (Table 1) where instead of random masking, we use the top paramters extracted by Algorithm 1 to show that as far as the sparisty ratio is considered, there is no different between specific targeting of parameters and random masking and in fact, random masking performs slightly better averaged over datasets empirically proving our theoretical insights. Therefore, our work *is adaptive* and *is dependent on the task* since we use a few-shot subset of the data to extract the sparsity ratios and masks.
> >  - **Q2 - Sparsity Ratio Derivation and Consistency**: As shown in Figure 13 in Appendix E, over different number of shots, there is a large overlap between the extracted subnetwork from both SVD-based and SNIP (gradient-based) methods. Therefore, considering that one of the main messages of our work is about the importance of sparsity ratios, the insignificant changes in the top indices across different shots shows that the sparsity ratio is not dependent on the variability of the shots. We'll add a discussion on this topic to the main text of the paper.
> >  - **Q3 - Layer-Wise Performance Variability**: Again, note that the sparsity in our method is not uniform and is instead layer-dependent and adaptive. Nevertheless, we do provide the results for an experiment with uniform sparsity ratio in Figure 5 with the description in Section 5.3. We also provide visualizations of subnetwork sizes in Figure 14, Figure 17 and layer-based magnitude of residuals in Figure 7, Figure 8, Figure 9, and Figure 10. We show that some layers do indeed require more modification and require more number of parameters compared to other layers. Pruning entire layers however is out of the scope of this work.
> >  - **Q4 - Comparison to suggested Works**: Thanks for suggestions. These are concurrent related works highlighting the signicant direction of our research that we will discuss in the related work. However, we note that [a] focuses on multi-task settings and task arithmetic that is not directly comparable. [b] is designed to strictly target language models and is concerned with multi-language tasks. We are running experiments on LLAMA2-7b as a larger model and will report the results when completed and compare with [c]. We will also add [c] to our related work.
> >  - **Q5 - Model Transferability**: Thanks for the suggestion. While we appreciate the suggestion, testing this would be out of the scope of the paper. We're pruning a LoRA to train on a specific dataset and show that sparsification of adapters works when using our approach. Therefore, our work masks the model based on a specific dataset and investigating transfer of this mask to another dataset would be out of the scope of our work.
> >
> > [a] Panda, Ashwinee, et al. "Lottery ticket adaptation: Mitigating destructive interference in LLMs." arXiv preprint arXiv:2406.16797 (2024).
> > [b] Xu, Jing, and Jingzhao Zhang. "Random Masking Finds Winning Tickets for Parameter Efficient Fine-tuning." arXiv preprint arXiv:2405.02596 (2024).
> > [c] Yuan, Fei, et al. "KS-Lottery: Finding Certified Lottery Tickets for Multilingual Language Models." arXiv preprint arXiv:2402.02801 (2024).

---

> > > ### Author Response · Authors · 2024-11-26
> > >
> > > **Missing comparison with existing Lottery Ticket-based LoRA methods**: We provide the results on LLAMA2-7b for our work compared against [a] here and have added a new section in the revised version of the main text. We thank the reviewer for suggesting this recent work.
> > >
> > > | Task   | FT            | LoRA           |  Masking (0.001%) | Partial-LoRA (0.001%) |
> > > |--------|---------------|----------------|-------------------|-----------------------|
> > > | SST-2  | 94.7(1e-6)    | 95.4(1e-4)     |  95.5(5e-2)       | 94.9(5e-2)            |
> > >
> > > Here we compare the results of our work against full fine-tuning, low-rank adaptation, and Balanced Masking techniques [e]. The related work section of the main text has also been revised accordingly. As evident in the table above, for LLAMA2-7b on SST-2, the sparsification of LoRA is also possible with larger models. We will be adding more datasets to this table in the coming few days. Compared to Balanced Masking, we obtain similar performance without the requirement of an expensive grid search over the sparsity ratio since we use the sparsity ratio derivation technique proposed in Section 4.2 of our paper.
> > >
> > > We briefly bring a comparison to [a] here that can also be found in our related work section. Our approach employs an efficient masking technique that is non-ad-hoc with respect to how the residuals are masked. We theoretically justify our method using the literature on LTH where [a] does not. Perhaps more importantly, we employ a forward-only mechanism adaptive to the required capacity by the task and derive the required sparsity used for sampling our random masks. [a] does not provide any approach to deriving the probability of the Bernoulli distribution for the random masks and for experimentation, performs a grid search on the sparsity ratio as a hyperparameter.
> > >
> > > These results have also been added to the revised manuscript in Section 5.4 and the new Table 1. Additionally, we will revise to include proposed changes for Q2 and Q7 shortly.
> > >
> > > [a] Xu, Jing, and Jingzhao Zhang. "Random Masking Finds Winning Tickets for Parameter Efficient Fine-tuning." arXiv preprint arXiv:2405.02596 (2024).

---

> > > > ### Author Response · Authors · 2024-11-29
> > > >
> > > > **Update on LLAMA2 results**:
> > > > We provide here the results of LLAMA2-7b on Copa and WSC as two additional datasets.
> > > >
> > > > | Task   | FT            | LoRA           |  Masking          | Partial-LoRA          |
> > > > |--------|---------------|----------------|-------------------|-----------------------|
> > > > | SST-2  | 94.7(6.7B)    | 95.4(4.2M)     |  95.5(0.068M)     | 94.9(0.068M)          |
> > > > | WSC    | 70.2(6.7B)    | 70.2(4.2M)     |  64.52(0.7M)      | 66.4(0.7M)            |
> > > > | Copa   | 87.0(6.7B)    | 85.0(4.2M)     |  88.0(0.068M)     | 84.0(0.060M)          |
> > > >
> > > > Here is a brief description of the two new tasks:
> > > >  - WSC: The Winograd Schema Challenge (WSC) dataset tests pronoun resolution by providing sentences with ambiguous pronouns, where the task is to identify the correct referent using commonsense reasoning.
> > > >  - Copa: The Choice of Plausible Alternatives (COPA) dataset consists of 1,000 premises, each with two possible alternatives, where the task is to determine which alternative is more plausible as a cause or effect of the premise.
> > > >
> > > > WSC Dataset Example:
> > > >  - Sentence: The trophy doesn’t fit in the suitcase because it is too large.
> > > >  - Question: What does "it" refer to?
> > > >  - Answer: The trophy.
> > > >
> > > > COPA Dataset Example:
> > > >  - Premise: The vase fell off the shelf.
> > > >  - Choice 1: It was hit by a ball.
> > > >  - Choice 2: It was on sale.
> > > >  - Correct Answer: Choice 1 (cause-effect relationship).
> > > >
> > > > On the Copa dataset, our approach approximates LoRAs with even fewer number of parameters which is determined by our subnetwork extraction method alongside the 90% margin. On the WSC dataset, our subnetwork extraction asks for more trainable parameters compared to the other two tasks reflecting the difficulty of this commonsense reasoning task leading to a slight performance decrease with pruning. Although due to the adaptive nature of our approach, we do obtain a higher EMA compared to [a].
> > > >
> > > > We hope that our revisions, clarifications, and the inclusion of new experiments sufficiently address the reviewer’s concerns and provide a basis to reconsider the score.
> > > >
> > > > [a] Xu, Jing, and Jingzhao Zhang. "Random Masking Finds Winning Tickets for Parameter Efficient Fine-tuning." arXiv preprint arXiv:2405.02596 (2024).

---

> ### Author Response · Authors · 2024-12-02
>
> **Summary of Our Rebuttal**
>
> We provide a very brief summary of our rebuttal as follows:
>
> **Adaptive Masking**:
> We clarified that our method is already naturally adaptive as a clear result of Algorithm 1.
>
> **Reproducibility**:
> We provide implementation details and the testing scenarios in Section 5. Our work sparsifies LoRAs and its variants. Therefore, we use the same hyperparameters as the work that our method is sparsifying and we have provided the details for the remaining hyperparameters in Section 5. We will add this to the implementation details in Section 5 and revise accordingly.
>
> **Layer-Wise Performance Variability**:
> We have provided extensive visualization and deep insights into the variations of the sparsity ratios across the layers (Figure 5, Figures 7 to 10, Figure 14, and Figure 17.).
>
> **Comparison to Suggested Works**: We are grateful for these suggestions and we provided our results on LLAMA2-7b on 3 datasets focusing on varying tasks and compared our results to the suggested work that was directly comparable to our work (Section 5.4 and Table 1).
>
> **Generalization Guarantees, Noise Testing, Adversarial Conditions, and Cross-Domain Transferability**:
> Our theoretical results backed by the empirical analysis on both vision and language tasks show that Partial-LoRAs can approximate LoRAs. Therefore, our work is focused on the approximation of LoRAs and masking low-rank adapters. Investigating generalization guarantees, adversarial conditions or cross-domain analysis would be out of the scope of this paper.
>
> We would be happy to elaborate further or clarify any of the points.

---

### Official Review · Reviewer_Fgsx · 2024-10-29

**Soundness:** 2
**Presentation:** 4
**Contribution:** 3
**Rating:** 6
**Confidence:** 4

**Summary:**

This paper proposes to reduce the parameters of LoRAs by only training a subset of rows/columns of the low-rank matrices. The feasibility of this approach is backed by a theoretical argument that extends the existence of sparse subnetworks in arbitrary models to models that include low-rank adapters. Experiments on six image classification tasks and four sentence classification tasks show that performance similar or better than the original LoRA formulation can be obtained with the proposed method. Additionally, this paper proposes a method to select the sparsity ratio and which rows/columns to prune based on relevant prior work in the pruning domain. The main finding here is that choosing specific rows/columns of the low-rank matrices is unnecessary, and selecting these randomly is enough as long as the optimal sparsity ratio is respected.

**Strengths:**

1. This paper presents an interesting and valuable observation that LoRA adapters can be sparsified to reduce fine-tuning parameters further. If this holds in all cases, it is of great interest and usefulness to the community.
2. The proposed method is supported by a theoretical insight.
3. The paper includes extensive ablations to confirm that randomly selecting rows/columns to tune is a simple and strong method to sparsify LoRAs. While this does not rule out that some method exists to strategically select these rows/columns, it should be encouraged to also report such negative insights.
4. The presentation of the paper is excellent, it is clear and well-written

**Weaknesses:**

While the proposed idea is interesting, the experimental evaluation should be improved:
1. Experiments and Algorithm 1 only consider classification tasks. However, other tasks, such as LLM instruction tuning, are also very relevant applications. This paper would greatly benefit from including at least one generation task in the evaluation and showing the applicability of the proposed method there as well.
2. Experiments are only conducted on a subset of the VTAB and GLUE benchmarks. However, no details are given on why these particular tasks were chosen and why the whole benchmarks are not evaluated.
3. It is unclear why results are only reported on the development split of GLUE tasks, although test set results can be obtained from the web API [1]. Furthermore, it is unclear if the GLUE development split was also used to select the top 5 hyperparameter configurations or if a different subset of the data was used.

[1] https://gluebenchmark.com/submit

**Questions:**

Overall, I think this paper misses the aspect that LoRA is not only used in classification settings. Thus, I recommend to add a variant of Algorithm 1 and experiments for natural language generation tasks such as instruction tuning. Furthermore, the evaluation should be more aligned with the original evaluation of baselines, for example in DoRA [1] or VeRA [2] by including full benchmarks and similar tasks. This would allow for a better assessment of the respective strengths and weaknesses of the proposed method.

Otherwise, the paper should make it more clear that the proposed method is for classification scenarios only and revise the storyline accordingly.

[1] Liu et al.: DoRA: Weight-Decomposed Low-Rank Adaptation. In ICML (2024) -- https://arxiv.org/pdf/2402.09353

[2] Kopiczko et al.: VeRA: Vector-Based Random Matrix Adaptation. In ICLR (2024) -- https://arxiv.org/pdf/2310.11454

---

> ### Author Response · Authors · 2024-11-25
>
> Thank you for your detailed review and for highlighting the clarity and presentation quality of our work, as well as the potential impact of our findings on the community.
>
> - **W1, Q1 - Evaluation Tasks**: LLM instruction tuning and task alike for autoregressive models still focus on next word prediction which is inherently a classification task. We don't see any additional benefits of such evaluation changing anything for any other tasks. What we are trying to show is not a new method for adaptation but showing that the outputs of a pruned LoRA using our approach can approximate the outputs of a fully-paramterized LoRA. Nevertheless, we are running experiments on LLAMA2-7b as a larger model and will report the results when completed.
>
> - **W2 - Choice of Tasks**: We chose 2 smaller and 2 larger datasets from GLUE which is the commonly used approach to evaluation on GLUE in recent works [a]. We can supplement these results with other datasets in GLUE for the camera-ready submission. As for vision experiments, our target was not the VTAB dataset since our work is not focused on improving image classifcation. We experiment on a wide array of datasets with narrow-focused datasets up to general image classification datasets. This is to evaluate the performance of our approach on datasets of different label granularity and observe the impact of LoRA sparsification. This includes Pets, CIFAR-10, CIFAR-100, Flowers, FER2013, GTSRB, FGVC-Aircraft and ImageNet-1k. If the reviewer believes another dataset would derliver unexpected results or new information, we'd be thankful for them to point out the dataset. We will add the corresponding results to the camera-ready submission.
>
> - **W3 - GLUE Development Set**: Due to the rate limits on the website provided for benchmarking the test set of GLUE, the recent work [b] have chosen to evaluate on the development set and slice out a part of the training set for validation. We have followed the same approach for train/val/test split since using the GLUE benchmark website would not allow for direct comparison to the SoTA methods.
>
> [a] Ilharco, G., Ribeiro, M., Wortsman, M., Gururangan, S., Schmidt, L., Hajishirzi, H., & Farhadi, A. (2022). Editing Models with Task Arithmetic. ArXiv, abs/2212.04089.
> [b] Zhang, Q., Chen, M., Bukharin, A., He, P., Cheng, Y., Chen, W., & Zhao, T. (2023). Adaptive Budget Allocation for Parameter-Efficient Fine-Tuning. ArXiv, abs/2303.10512.

---

> > ### Comment · Reviewer_Fgsx · 2024-11-25
> > **Reviewer Response to Authors**
> >
> > Thank you very much for the answer to my review. Unfortunately, my main concerns have mostly not been adequately addressed.
> >
> > >  LLM instruction tuning and task alike for autoregressive models still focus on next word prediction which is inherently a classification task.
> >
> > I am not aware of language modeling being evaluated by accuracy. This may be true for some benchmarks which only evaluate one predicted token, but the proposed Algorithm 1 is not practical in free-generation settings. There, accuracy is not a viable metric.
> >
> > >  We don't see any additional benefits of such evaluation changing anything for any other tasks.
> >
> > Reviewers almost uniformly agree that additional evaluations, such as applying the proposed method to LLMs, would greatly improve the paper. In addition, a broader evaluation would greatly enhance comparability to other PEFT methods. The claim "the outputs of a pruned LoRA using our approach can approximate the outputs of a fully-parameterized LoRA" requires substantial empirical backing, ideally on commonly evaluated benchmarks (and not subsets thereof).
> >
> > >   * Nevertheless, we are running experiments on LLAMA2-7b as a larger model and will report the results when completed.
> > >   * We can supplement these results with other datasets in GLUE for the camera-ready submission
> >
> > I appreciate the effort in running additional large-scale experiments. Ideally, these should already be included in the rebuttal revision, especially because multiple reviewers requested such experiments. I am uncomfortable recommending a paper for acceptance where addressing main limitations is deferred to the camera-ready version. In general, I am curious why no rebuttal revision has been provided which includes at least some of the promised improvements.
> >
> > > Due to the rate limits on the website provided for benchmarking the test set of GLUE, the recent work [b] have chosen to evaluate on the development set and slice out a part of the training set for validation. We have followed the same approach for train/val/test split since using the GLUE benchmark website would not allow for direct comparison to the SoTA methods.
> >
> > Indeed it seems common practice to evaluate on the GLUE validation split, so the compatibility with previous work would be limited when using the test set.
> >
> > In summary, although I am still convinced by the utility and novelty of the proposed method, I think the evaluation needs to be broadened and the method adapted to free-text generation in order for this paper to be ready for publication. Thus, I am adapting my score accordingly.
> >
> > Of course, I am willing to raise the score if my concerns are adequately addressed, ideally in a revised pdf which can be uploaded during the discussion period.

---

> > > ### Author Response · Authors · 2024-11-26
> > >
> > > **Evaluation by Accuracy**: We used benchmarks that have accuracy as a main evaluation criteria to allow for a fair comparison to recently proposed approaches [a, b, c]. Open-ended or label-based, every benchmark requires a form of quantitative evaluation and that metric could simply replace the accuracy used in Algorithm 1. We provide a list of partially and completely open-ended benchmarks alongside their evaluation metric:
> > >  - TruthfulQA (Open-Ended): **Accuracy**
> > >  - ARC (Open-Ended): **Accuracy**
> > >  - SQuAD (Open-Ended): Exact Match **Accuracy** (EMA), F1 Score
> > >  - BigBench (Open-Ended): **Accuracy**, F1 Score, ROGUE, BLEU
> > >  - GSM8k (Partially Open-Ended): Exact Match **Accuracy**
> > >  - BBH (Partially Open-Ended): **Accuracy**, F1 Score
> > >  - MMLU (Multiple-Choice): **Accuracy**
> > >  - TydiQA (Partially Open-Ended): Exact Match **Accuracy**, F1 Score
> > >
> > > Clearly, the open-ended free-generation LLM benchmarks also use accuracy as an evaluation criteria. The work cited by the reviewer themselves [d] also uses variations of the above benchmarks which are evaluated using generation accuracy. Our response regarding the next token prediction was meant to convey that LLMs are commonly evaluated using criteria that can be used with the 90% margin proposed in our work (Section 4.2.1) and testing on such benchmarks would not change our methodology. If the reviewer is aware of any benchmarks where quantitative evaluation using the abovementioned metrics is impossible, we would appreciate it if they could share it with us for further examination.
> > >
> > > **Experiments and Evaluations**: We address the usage of LLMs in the next comment. We provide experiments for both vision and language modalities using ViT-B-16 and Deberta models. The vision experiments are done through a narrow to general image classification settings spanning 8 datasets. The language experiments are done on 2 smaller and 2 larger datasets from the GLUE benchmark. We have applied our approach to two low-rank based methods and compared to 4 state-of-the-art low-rank adaptation methods alongside LoRA (Figures 2, 3, and 4). All the evaluations on 12 total datasets across 280 training sessions are accompanied by comprehensive ablation studies highlighting the size of extracted subnetworks, difference in magnitude for residuals from the adapters compared to LoRA (Figures 7 to 10), studies on random subnetworks vs. targeted subnetworks (Table 1), alternative pruning approaches (Figure 5), and ablations on rank (Section 5.5).
> > >
> > > On the matter of GLUE subsets, it is common to evaluate only on a subset of the GLUE benchmark. The paper cited by the reviewer themselves [b] also uses a subset of the GLUE benchmark as opposed to the entire benchmark in Table 2.
> > >
> > > [a] Zhang, Qingru et al. “Adaptive Budget Allocation for Parameter-Efficient Fine-Tuning.” ArXiv abs/2303.10512 (2023): n. pag.
> > >
> > > [b] Kopiczko et al.: VeRA: Vector-Based Random Matrix Adaptation. In ICLR (2024) -- https://arxiv.org/pdf/2310.11454
> > >
> > > [c] Hu, J.E., Shen, Y., Wallis, P., Allen-Zhu, Z., Li, Y., Wang, S., & Chen, W. (2021). LoRA: Low-Rank Adaptation of Large Language Models. ArXiv, abs/2106.09685.
> > >
> > > [d] Liu et al.: DoRA: Weight-Decomposed Low-Rank Adaptation. In ICML (2024) -- https://arxiv.org/pdf/2402.09353

---

> ### Author Response · Authors · 2024-11-26
>
> **Additional LLM Results on LLama2**: We provide the results on LLAMA2-7b here and have added a new section in the revised version of the main text. The percentages inside parentheses show the fraction of the number of trainable parameters compared to that of full finetuning.
> | Task   | FT            | LoRA           |  Masking (0.001%) | Partial-LoRA (0.001%) |
> |--------|---------------|----------------|-------------------|-----------------------|
> | SST-2  | 94.7(1e-6)    | 95.4(1e-4)     |  95.5(5e-2)       | 94.9(5e-2)            |
>
> Here we compare the results of our work against full fine-tuning, low-rank adaptation, and Balanced Masking techniques [e]. We have added this new work as a comparison to other masking techniques for PEFT approaches recommended by reviewer vQUE. The related work section of the main text has also been revised accordingly. As evident in the table above, for LLAMA2-7b on SST-2, the sparsification of LoRA is also possible with larger models. We will be adding more datasets (starting with free-generation datasets) to this table in the coming few days. Compared to Balanced Masking, we obtain similar performance without the requirement of an expensive grid search over the sparsity ratio since we use the sparsity ratio derivation technique proposed in Section 4.2 of our paper.
>
> We briefly bring a comparison to [e] here that can also be found in our related work section. Our approach employs an efficient masking technique that is non-ad-hoc with respect to how the residuals are masked. We theoretically justify our method using the literature on LTH where [e] does not. Perhaps more importantly, we employ a forward-only mechanism adaptive to the required capacity by the task and derive the required sparsity used for sampling our random masks. [e] does not provide any approach to deriving the probability of the Bernoulli distribution for the random masks and for experimentation, performs a grid search on the sparsity ratio as a hyperparameter.
>
> We have also updated the submission with a revised version reflecting the discussed changes.
>
> [e] Xu, Jing, and Jingzhao Zhang. "Random Masking Finds Winning Tickets for Parameter Efficient Fine-tuning." arXiv preprint arXiv:2405.02596 (2024).

---

> ### Comment · Reviewer_Fgsx · 2024-11-26
> **Reviewer Response to Authors**
>
> I want to thank the authors for their continued efforts in providing additional clarifications and experiments.
>
> > Open-ended or label-based, every benchmark requires a form of quantitative evaluation and that metric could simply replace the accuracy used in Algorithm 1.
>
> Thank you for the clarification. I think this aspect should be stressed in the paper, perhaps alongside a discussion on the factor 0.9 as threshold. This factor appears quite intuitive in classification settings, but is the same true for all relevant metrics (e.g. CIDEr, Rouge, BLEU, F1, ...)? Another case to consider is when the base model (before fine-tuning) achieves 0 accuracy on the task at hand or cannot perform it at all for different reasons. This case is relevant, as LoRA is not dependent on any prior performance.
>
> >   * We provide a list of partially and completely open-ended benchmarks alongside their evaluation metric
> >   * If the reviewer is aware of any benchmarks where quantitative evaluation using the abovementioned metrics is impossible, we would appreciate it if they could share it with us for further examination.
>
> Clearly, many common benchmarks use accuracy and similar classification metrics for evaluation, and the proposed method is applicable there. One type of evaluation not considered seems to be LLM-as-a-judge such as applied to MT-Bench [1] LLaVA [2] or DPO [3], where performance is relative to another model or ground truth text answers. However, it seems possible to handle this in the same way as suggested for other quantitative metrics as suggested in the previous response.
>
> A general aspect to consider is that fine-tuning is a very general framework that can be applied to both supervised and unsupervised tasks in greatly varying settings. The main application of PEFT methods are not benchmarks, even if we evaluate the methods there and expect that the observed performance generalizes. If a method with the same generality as LoRA is proposed, care must be taken not to restrict the scope of application, or clarify what is the intended (narrower) scope of application.
>
> >  The vision experiments are done through a narrow to general image classification settings spanning 8 datasets. The language experiments are done on 2 smaller and 2 larger datasets from the GLUE benchmark.
>
> Compared to DoRA [4] and VeRA [5], the number of different tasks is still relatively small, even if the number of datasets is comparable. For example, DoRA features 4 tasks (treating video and image understanding as separate tasks) and VeRA features  4 tasks as well. I understand that the main comparison target of this work is LoRA, but the evaluation presented there is not replicated, and then it becomes difficult to understand why these particular tasks are chosen. Without common grounds, my worry is that tasks can be cherry-picked from a relatively large pool of possible applications.
>
> > We provide the results on LLAMA2-7b here and have added a new section in the revised version of the main text
>
> I appreciate the effort in providing these additional experiments, and I also appreciate that a revised pdf has been made available. It would have been great to see results on an additional task (instead of a new model on a GLUE task which had already been evaluated before). Finally, why does full fine-tuning only tune a fraction of 1e-6 parameters of the model?
>
> In summary, I would like to ask the authors to add a discussion on the metric in Algorithm 1 where they explain how the proposed method can be used for arbitrary tasks and metrics. If also additional tasks can be added to the evaluation, I think the paper will be much stronger and I will be happy to further raise my score.
>
> [1] Zheng et al.: Judging LLM-as-a-Judge with MT-Bench and Chatbot Arena. In NeurIPS, 2023\
> [2] Liu et al.: Visual Instruction Tuning. In NeurIPS, 2023\
> [3] Rafailov et al.: Direct Preference Optimization: Your Language Model is Secretly a Reward Model. In NeurIPS, 2023\
> [4] Liu et al.: DoRA: Weight-Decomposed Low-Rank Adaptation. In ICML, 2024\
> [5] Kopiczko et al.: VeRA: Vector-Based Random Matrix Adaptation. In ICLR, 2024

---

> > ### Author Response · Authors · 2024-11-29
> >
> > **Usefulness of other metrics for Algorithm 1** We will revise to emphasize the possibility of using metrics other than accuracy. We refer to the 4th point in this comment and the next comment for the usage of other metrics.
> >
> > **Cases where base model achieves 0 accuracy**: Low-rank adaptation is meant for modifying a pre-trained model by adding a low-rank update to its weight matrix, without drastically altering the model's overall structure [a]. Studies have shown that LoRA does not significantly modify the model's parameters, and the low-rank update is mainly targeted at the pre-trained model [a]. Therefore, in such cases, LoRAs are not expected to be used. We will revise our discussion section to elaborate on this point.
> >
> > **Scope of application**: We agree that in a real-world settings, LoRAs may be used in other tasks dissimilar to benchmarks. However, benchmarks allow for an objective and controlled examination of the methodology proposed since experimentation on real-world settings would not allow for comparisons to the SotA approaches in a fair setting.
> >
> > **Number of Datasets**: While we sought to provide extensive empirical evaluations, we understand more experiments could improve the presentation. We provide our results on two more GLUE datasets in the following table with their corresponding number of parameters. Please refer to the next point for another two tasks on LLMs (adding to four new tasks).
> > | Method                     | QQP (Accuracy/F1)     | MNLI    (Matched/Mismatched)  |
> > |----------------------------|-----------------------|-------------------------------|
> > | AdaLoRA                    | 92.23/89.74 (1.32M)   | 90.36/90.43 (1.32M)           |
> > | Partial-AdaLoRA            | 91.93/89.36 (0.45M)   | 90.10/90.00 (0.24M)           |
> >
> > Here we specify our reasoning for choosing these datasets: QQP is evaluated using both accuracy and the F1 metric. To extract the sparsity ratios, we used F1 as opposed to accuracy as an empirical way of confirming the first point in this comment. As evident, our approach also works when using F1 and the performance does not degrade. MNLI is evaluated on in-domain and out-of-domain genres called matched and mismatched splits meant to evaluate domain robustness. Again, the performance does not degrade with our approach. We will add these results to the revised manuscript.
> >
> > Please note that we are not proposing a completely new adaptation approach. Instead, we sparsify LoRAs and as such have directly compared with those approaches. We show the effectiveness of our work and how this sparsity affects various aspects of LoRA residuals. Importantly, we, for the first time, show lottery ticket in low-rank adapters.
> >
> > [a] Shuttleworth, R., Andreas, J., Torralba, A., & Sharma, P. (2024). LoRA vs Full Fine-tuning: An Illusion of Equivalence.

---

> > > ### Author Response · Authors · 2024-11-29
> > >
> > > **LLAMA2 results**:
> > > We provide here the results of LLAMA2-7b on the Copa and WSC datasets.
> > >
> > > | Task   | FT            | LoRA           |  Masking          | Partial-LoRA          |
> > > |--------|---------------|----------------|-------------------|-----------------------|
> > > | SST-2  | 94.7(6.7B)    | 95.4(4.2M)     |  95.5(0.068M)     | 94.9(0.068M)          |
> > > | WSC    | 70.2(6.7B)    | 70.2(4.2M)     |  64.52(0.7M)      | 66.4(0.7M)            |
> > > | Copa   | 87.0(6.7B)    | 85.0(4.2M)     |  88.0(0.068M)     | 84.0(0.060M)          |
> > >
> > > Here is a brief description of the two new tasks:
> > >  - WSC: The Winograd Schema Challenge (WSC) dataset tests pronoun resolution by providing sentences with ambiguous pronouns, where the task is to identify the correct referent using commonsense reasoning.
> > >  - Copa: The Choice of Plausible Alternatives (COPA) dataset consists of 1,000 premises, each with two possible alternatives, where the task is to determine which alternative is more plausible as a cause or effect of the premise.
> > >
> > > WSC Dataset Example:
> > >  - Sentence: The trophy doesn’t fit in the suitcase because it is too large.
> > >  - Question: What does "it" refer to?
> > >  - Answer: The trophy.
> > >
> > > COPA Dataset Example:
> > >  - Premise: The vase fell off the shelf.
> > >  - Choice 1: It was hit by a ball.
> > >  - Choice 2: It was on sale.
> > >  - Correct Answer: Choice 1 (cause-effect relationship).
> > >
> > > On the Copa dataset, our approach approximates LoRAs with even fewer number of parameters which is determined by our subnetwork extraction method alongside the 90% margin. On the WSC dataset, our subnetwork extraction asks for more trainable parameters compared to the other two tasks reflecting the difficulty of this commonsense reasoning task leading to a slight performance decrease with pruning. Although due to the adaptive nature of our approach, we do obtain a higher EMA compared to [a].
> > >
> > > **Number of full-finetuning parameters**:
> > > We apologize regarding the inconsistency in notation of the table in the previous comment. The values inside the parentheses represent the learning rate used for finetuning as opposed to our other tables where these values represented the number of parameters. The modified table in the previous point is consistent with the structure of other tables.
> > >
> > > While the approach we compare with performs grid search over every hyperparameter, we chose to report averaged results of 3 training sessions with different seeds all with the same learning rate (5e-2). For full fine-tuning, LoRA, and Masking [a], learning rates of 1e-6, 1e-4, and 5e-2 were used, respectively.
> > >
> > > [a] Xu, Jing, and Jingzhao Zhang. "Random Masking Finds Winning Tickets for Parameter Efficient Fine-tuning." arXiv preprint arXiv:2405.02596 (2024).

---

> > > > ### Comment · Reviewer_Fgsx · 2024-11-29
> > > > **Reviewer Response to Authors**
> > > >
> > > > Thank you very much for the detailed answers. With the latest response, most of my concerns have been addressed. In particular,
> > > >   * Clarifications have been provided regarding Algorithm 1, and they will be included in the camera-ready version
> > > >   * Evidence has been provided that shows that the proposed method also generalizes to more tasks than considered in the original submission. These results will be included in the camera-ready version
> > > >
> > > > I appreciate the authors' continuing willingness to engage in discussion and provide additional clarification. With the proposed and promised modifications, I find the method of sufficient interest and also sufficiently supported by empirical results so it can be considered for publication. I am thus raising my score.

---

### Official Review · Reviewer_44af · 2024-11-02

**Soundness:** 4
**Presentation:** 3
**Contribution:** 3
**Rating:** 6
**Confidence:** 4

**Summary:**

The paper explores the concept of "winning tickets" within the Low-Rank Adaptation, inspired by the Lottery Ticket Hypothesis. The authors present a new method (denoted Partial-LoRA) that relies on random masking to identify sparse low-rank networks that can maintain or improve performance in fine-tuning pre-trained models on specific tasks while trying to reduce the number of trainable parameters. The authors present experiments on various vision and language tasks demonstrating significant parameter reductions while maintaining or improving accuracy.

**Strengths:**

1. The paper proposes a novel, interesting method that extends the lottery ticket hypothesis to low-rank adaptation.
2. The authors provide a good theoretical foundation to justify the LTH concept within LoRA.
3. The experiments cover a wide range of vision and language tasks, providing solid results in terms of maintaining accuracy while reducing the total number of parameters.

**Weaknesses:**

1. I think there is a main weakness in the presentation of the results: while the plots are nice, there should be a table showing the number of parameters and accuracy. Additionally, it should compare against methods of close parameter count if feasible (e.g., current results show how the method can maintain the accuracy while using few param counts, there should be a comparison the other way around, too, showing how methods with similar parameter count cannot achieve the same result to justify the original methods weren’t over -parameterized to being with).
2. Lack of experiments on larger models (e.g., llama), which are usually the main beneficiaries of PEFT.
3. The approach's reliance on random masking, rather than more targeted sparsification, could potentially limit its performance on more complex tasks or architectures.
4. The method relies on random masking for the LoRA matrices, which is itself low-rank (compressed), i.e., the mask itself is practically compressed; the method could benefit from comparing masking full-rank weights vs low-rank adapter.
5. A separate training phase on a subset dataset is needed to identify sparsity ratios.
6. Minor nit, the theorems can be a bit more rigorous (e.g., define all symbols T, U..etc, instead of relying on the context or the common use).

**Questions:**

1. As I mentioned in the weakness, is there a comparison with methods with similar parameter counts showing that your method can achieve x improvement using y params, while other methods can’t achieve the same improvement using also ~y params?
2. Why do masked LoRAs sometimes significantly outperform full LoRAs (e.g., Flowers dataset)? Is this purely preventing overfitting, or is there a more fundamental advantage?
3. The approach randomly prunes weights without considering inter-layer dependencies. Could more intelligent layer-wise tuning (e.g., adjusting pruning intensity per layer based on importance) improve results?
4. Did you experiment with larger models (e.g. LLM) that usually benefit from PEFT more? Or is there a reason for sticking to smaller models?

---

> ### Author Response · Authors · 2024-11-25
>
> Thank you for your insightful review and for appreciating the novelty and theoretical foundation of our work.
>
> Due to space constraints, we'll address all the points in the weaknesses section here:
> - **W1 - Table with All Results**:Thanks for your suggestion. We'll add a table in the Appendix quantitatively detailing the parameter count alongside the accuracy for each experiment.
> - **W1, Q1 - Reducing the parameters on the full adapter**: The only way to reduce the number of parameters of the fully-parameterized adapters would be to reduce their rank which would be unfair since then they would have fewer subspaces accessible during finetuning. Nevertheless, Figure 6 in Section 5.5 allows for comparison of different ranks for LoRA to that of Partial-LoRA where the fewer ranks of LoRA cannot achieve the same accuracy as Partial-LoRA with a higher rank. We will add a secondary x-axis to Figure 6 showing the number of parameters, allowing for parameter and rank comparison simultaneously.
> - **W2 - Larger Models**: We are running experiments on LLAMA2-7b as a larger model and will report the results when completed.
> - **W3 - Targeted Sparsification**: We provide results for such a method we named "Targeted-LoRAs" in Table 1 of the main text where instead of using the top extracted parameters only for sparsity ratio, we target those parameters specifically. We show in Table 1 that Targeted-LoRAs perform worse than Partial-LoRAs on average and this slightly lower performance from Targeted-LoRA remains consistent across different number of shots as shown in Table 2 in Appendix B. Additionally, we provide an inbetween method where we use the extracted top parameters but treat them stochastically and provide those results in Figure 11 in Appendix D to show the potential of introducing randomness to the targeting process.
> - **W4 - Masking Model vs. Masking LoRAs**: There is a threshold for sparsity after which the number of parameters required for training when the model itself is masked is less than the same number when using LoRAs: A LoRA has  $m \times d + d \times n$ parameters, where $m$ and $n$ are the dimensions of the model's weight matrix, and $d$ is the LoRA's rank. To determine the threshold where training the full model is better than using a masked LoRA, we compare the number of parameters where $m \times n \leq m \times d + d \times n$. Simplify to get $\frac{m \times n}{m + n} \leq d$. Therefore, this threshold is dependent on the rank of the LoRA. For example, considering ViT-B-16, we have matrices of size $3072 \times 768$, in a simplified case of balanced pruning on rows and columns, the mask would have to sparsify the rows and columns to only 25 rows and 5 columns for masking the model itself to be more efficient than a LoRA with a rank of 4. Therefore, the number of parameters for such as scenario would be very small unless the weight matrix itself is small to begin with. In such a case where the model is small, a LoRA would already not be appropriate for fine-tuning.
> - **W5 - Necessity for a Subset Dataset for Identifying Sparsity Ratios**: For training LoRAs we will always need a dataset. We have shown that a few-shot dataset is enough to extract masks (lines 312-314). The same dataset is used for training the subnetwork. Note that for open-world prediction models such as CLIP, a separate training phase is not necessary since we can extract the subnetwork by initializing the linear classifier using the text encoder as discussed in Section 5 under Datasets and Models.
> - **W6 - Revision of Theorem to Make it More Rigorous**: We will revise the theorem to make it more rigorous.

---

> > ### Author Response · Authors · 2024-11-25
> >
> > Here we address the questions section:
> > - **Q2 - Masked LoRA Outperforming LoRA**: As mentioned in Section 5.2, we believe this is due to the overparameterization of other methods causing overfitting since Flowers has a much smaller training set compared to the other datasets.
> > - **Q3 - Inter-Layer Dependencies**: With our approach, watching out for inter-layer dependencies is not necessary. We provide results in Section H, Table 3 where we score parameters based on preservation of the flow from one layer to another to prevent zeroing out information flow. In this table, it is evident that using a continous method by watching out for flow preservation has no effects on the performance of the model. We attribute this to how our approach is concerned with pruning adapters that use residual weights on top of a pretrained model and therefore, by zeroing out a connection, the same connection inside the pre-trained model remains the same and the flow of information in the model does not get cut with pruning the adapter. Please refer to Section 4.1 for more details. To prevent ambiguity, please note that we already do adjust pruning intensity per layer based on importance of elements using the sparsity ratio derived in Section 4.2.
> > - **Q4 - Model Size**: We use ViT-B-16 for vision and Deberta-v3-base for language models which are the commonly used benchmarking models for works on LoRAs to be able to directly compare our work to the literature. Interestingly, smaller models are harder to work with according to the literature [a]. We are also running experiments on LLAMA2-7b as a larger model and will report the results when completed.
> > [a] Advait Gadhikar, Sohom Mukherjee, and Rebekka Burkholz. Why Random Pruning Is All We Need to Start Sparse, May 2023. URL http://arxiv.org/abs/2210.02412. arXiv:2210.02412 [cs].

---

> > > ### Author Response · Authors · 2024-11-26
> > >
> > > **Table with All Results**: We have added a table for each modality reporting the quantitative results in more detail alongside the exact parameter count in the Appendix I alongside a reference to this Appendix section in Section 5.2.
> > >
> > > **Reducing the parameters on the full adapter**: We have updated the submission manuscript to reflect the discussed changes in Figure 6.
> > >
> > > **Larger Models**: We provide the results on LLAMA2-7b here and have added a new section in the revised version of the main text.
> > > | Task   | FT            | LoRA           |  Masking (0.001%) | Partial-LoRA (0.001%) |
> > > |--------|---------------|----------------|-------------------|-----------------------|
> > > | SST-2  | 94.7(1e-6)    | 95.4(1e-4)     |  95.5(5e-2)       | 94.9(5e-2)            |
> > >
> > > Here we compare the results of our work against full fine-tuning, low-rank adaptation, and Balanced Masking techniques [e]. We have added this new work as a comparison to other masking techniques for PEFT approaches recommended by reviewer vQUE. The related work section of the main text has also been revised accordingly. As evident in the table above, for LLAMA2-7b on SST-2, the sparsification of LoRA is also possible with larger models. We will be adding more datasets to this table in the coming few days. Compared to Balanced Masking, we obtain similar performance without the requirement of an expensive grid search over the sparsity ratio since we use the sparsity ratio derivation technique proposed in Section 4.2 of our paper.
> > >
> > > We briefly bring a comparison to [a] here that can also be found in our related work section. Our approach employs an efficient masking technique that is non-ad-hoc with respect to how the residuals are masked. We theoretically justify our method using the literature on LTH where [a] does not. Perhaps more importantly, we employ a forward-only mechanism adaptive to the required capacity by the task and derive the required sparsity used for sampling our random masks. [a] does not provide any approach to deriving the probability of the Bernoulli distribution for the random masks and for experimentation, performs a grid search on the sparsity ratio as a hyperparameter.
> > >
> > > These results have also been added to the revised manuscript in Section 5.4 and the new Table 1. Additionally, we will revise Theorem 4.1 before the discussion period ends.
> > >
> > > [a] Xu, Jing, and Jingzhao Zhang. "Random Masking Finds Winning Tickets for Parameter Efficient Fine-tuning." arXiv preprint arXiv:2405.02596 (2024).

---

> > > > ### Comment · Reviewer_44af · 2024-11-26
> > > >
> > > > Dear authors,
> > > > Thank you for the detailed responses and showing the results on Llama2-7b (here and in the main text), I think this table should be clearer in the final version, the rest of the questions have been answered.

---

### Official Review · Reviewer_H8Ln · 2024-11-04

**Soundness:** 1
**Presentation:** 2
**Contribution:** 1
**Rating:** 3
**Confidence:** 4

**Summary:**

This paper investigates the existence of sparse LoRA adapters in the context of the Lottery Ticket Hypothesis (LTH). The authors give some theoretical proof that sparse subnetworks exist for low-rank adapters. They also propose a method, Partial-LoRA, to add sparse low-rank parameters to pre-trained models. The experiments show that Partial-LoRA can reduce trainable parameters by up to 87% while maintaining or even improving model performance in some cases.

**Strengths:**

The authors address an interesting problem and draw inspiration from previous works. The paper is well-organized and structured.

**Weaknesses:**

Please refer to the questions I listed.

**Questions:**

1. The theoretical proof in this paper, specifically **Theorem 4.1**, is based on previous works, such as **Theorem 2.5** in (Gadhikar et al., 2023), which proves the existence of a sparse mask satisfying the error bound. However, the authors make an ambiguous statement in Theorem 4.1, not clarifying whether they are discussing the existence (there exists a mask U that satisfies the condition) or universality (for all mask U, the condition is satisfied). This needs to be clarified.
2. Following above question, if Theorem 4.1 is discussing the existence of a sparse mask, how to ensure the mask satisfying Theorem 4.1 can be decomposed into a low-rank format as in Equation (6)? Otherwise, if Theorem 4.1 is discussing the universality of the sparse mask, the proof provided by the authors is not sufficient because the reference (Gadhikar et al., 2023) only proves the existence of a sparse mask.
3. The experiments in this paper are not so convincing in 2024, such as the DeBERTa-v3-base (86M) model and GLUE benchmark. The authors should consider using more recent models and benchmarks to evaluate the performance of Partial-LoRA.
4. The most significant impact of Partial-LoRA is reducing the number of trainable parameters. However, the authors do not provide any analysis and evidence on why we should care about this metric. Considering that the LoRA adapters do not account for a significant portion of the model size and computation, the authors should provide more insights into the importance of pruning them.

---

> ### Author Response · Authors · 2024-11-25
>
> Thank you for your thoughtful review and for highlighting both the strengths and areas for improvement in our work.
>
> - **Q1, Q2 - Clarification on the Mask and Theorem 4.1**: We explicitly state that *there exists* a sparse mask $U$ in line 222 where after the application of this mask to the residuals of a low-rank adapter, the outputs of this adapter stay within a bound compared to the outputs of an unmasked adapter. This point is more intuitively discussed in lines 222 to 226. Therefore, our proof is sufficient in the context of our work and as an extention of [a]. We will revise and clarify further in the main therorem text. We also note that we have clearly cited and shown the link between our work and Theorem 2.5 from [a]: there is no proof for the bound for adapters in this prior work. Moreover, $U$ is only required to be sparse and the approach we took to decompose it has no implications in the theorem. However, our approach leads to much more efficient implementation. In other words, there is no computation necessary for the decomposition of the mask as explained in lines 289 to 294.
>
> - **Q3 - Evaluation on recent models**: We use Deberta for our language experiments and ViT-B-16 for our vision experiments. The reason for choosing these models was to directly compare with state-of-the-art [b]. The current literature commonly uses Deberta alongside the GLUE benchmark due to how streamlined the comparison can be to the recent works. We're not sure what the reviewer thinks new experiments on newer models would reveal that isn't already addressed by the current experiments, given that the structure of these models is identical to DeBERTa. The recency of the recent models does not introduce an aspect of finetuning that is different to that of less recent models. Nevertheless, we are running experiments on LLAMA2-7b as a larger model and will report the results when completed.
>
> - **Q4 - Significance of the Parameter Reduction** We first would like to highlight that the significance of our work is in drawing connection between the low-rank adapter models and lottery ticket hypothesis that has never been investigated before. Based on this theoretical insight, we designed an algorithm that shows the parameter size could be in fact even lower in the adapters that will even lead to higher performance. We hope the merits of our work is seen in this light. Second, a practical impact of our lower paramter count is in the case of models that do not fit on consumer hardware. For example, in cases such as LLAMA-7b, without memory reduction techniques, finetuning is impossible on consumer hardware. Given issues arising with finetuning with quantization [c], our approach enables finetuning without relying on such hardware oriented techniques, allowing for democratization of fine-tuning.
>
> [a] Advait Gadhikar, Sohom Mukherjee, and Rebekka Burkholz. Why Random Pruning Is All We Need to Start Sparse, May 2023. URL http://arxiv.org/abs/2210.02412. arXiv:2210.02412 [cs].
>
> [b] Zhang, Qingru et al. “Adaptive Budget Allocation for Parameter-Efficient Fine-Tuning.” ArXiv abs/2303.10512 (2023): n. pag.
>
> [c] Gholami, A., Kim, S., Dong, Z., Yao, Z., Mahoney, M. W., & Keutzer, K. (2021). A survey of quantization methods for efficient neural network inference. CoRR, abs/2103.13630. https://arxiv.org/abs/2103.13630

---

> > ### Author Response · Authors · 2024-11-26
> >
> > **Evaluation on recent models**: We provide the results on LLAMA2-7b here and have added a new section in the revised version of the main text.
> > | Task   | FT            | LoRA           |  Masking (0.001%) | Partial-LoRA (0.001%) |
> > |--------|---------------|----------------|-------------------|-----------------------|
> > | SST-2  | 94.7(1e-6)    | 95.4(1e-4)     |  95.5(5e-2)       | 94.9(5e-2)            |
> >
> > Here we compare the results of our work against full fine-tuning, low-rank adaptation, and Balanced Masking techniques [e]. We have added this new work as a comparison to other masking techniques for PEFT approaches recommended by reviewer vQUE. The related work section of the main text has also been revised accordingly. As evident in the table above, for LLAMA2-7b on SST-2, the sparsification of LoRA is also possible with larger models. We will be adding more datasets to this table in the coming few days. Compared to Balanced Masking, we obtain similar performance without the requirement of an expensive grid search over the sparsity ratio since we use the sparsity ratio derivation technique proposed in Section 4.2 of our paper.
> >
> > We briefly bring a comparison to [a] here that can also be found in our related work section. Our approach employs an efficient masking technique that is non-ad-hoc with respect to how the residuals are masked. We theoretically justify our method using the literature on LTH where [a] does not. Perhaps more importantly, we employ a forward-only mechanism adaptive to the required capacity by the task and derive the required sparsity used for sampling our random masks. [a] does not provide any approach to deriving the probability of the Bernoulli distribution for the random masks and for experimentation, performs a grid search on the sparsity ratio as a hyperparameter.
> >
> > These results have also been added to the revised manuscript in Section 5.4 and the new Table 1.
> >
> > [a] Xu, Jing, and Jingzhao Zhang. "Random Masking Finds Winning Tickets for Parameter Efficient Fine-tuning." arXiv preprint arXiv:2405.02596 (2024).

---

> ### Comment · Reviewer_H8Ln · 2024-11-27
>
> I extend my gratitude to the authors for their comprehensive response, and here are some of my follow-up questions:
>
> - I appreciate the authors' commitment to refining the statement of Theorem 4.1. A statement following the style similar to theorem 2.5 in (Gadhikar et al., 2023), which explicitly states "*Then with probability ..., there exists a mask so that ...*", would greatly enhance the clarity and understanding of the theorem.
>
> - I remain concerned about the connection between the proposed method and lottery ticket hypothesis as claimed in the paper. Theorem 4.1 is proving the existence of winning ticket, represented by a sparse mask $U$. Specifically, Theorem 4.1 does not address the existence of a winning ticket that can be written in the format of Equation (6). It is unclear to me how the sparse LoRA with $u_{row}$ and $u_{col}$ by Equation (6) can be guaranteed to be a winning ticket. Could the authors provide more details and explanation on this?
>
> - I appreciate the authors' experimental results with LLAMA-7b. In addition to the accuracy results, could the authors also provide data on memory reduction? This would help demonstrate the effectiveness of the proposed method in enabling fine-tuning without relying on hardware-specific techniques, as claimed.

---

> > ### Comment · Reviewer_H8Ln · 2024-11-28
> > **following question about memory reduction**
> >
> > Particularly, here is a very rough estimation of LoRA's memory footprint for LLAMA-2-7b:
> > | Argument | Value |
> > | --- | --- |
> > | Memory for Base Model | 7B * 2 (fp16) = 14 GBytes |
> > | Trainable Parameters | Since I couldn't find the exact number of trainable parameters in the paper, 0.042B $\approx$ 0.06% of 7B model (rank=8, Q,V only) is used as an estimation.  |
> > | Memory for each LoRA Parameter | (fp32, param) + (fp32, gradient), (fp32*2, AdamW states) = 16Bytes |
> > | Activation Memory | 2 GBytes |
> >
> > Thus, the GPU memory footprint of LoRA is estimated to be 0.042B * 16Bytes = 0.67 GBytes, which accounts only a very small portion of the total memory footprint (4%). So the memory reduction by sparsifyinng the LoRA is very limited, and the authors may want to provide more insights on this aspect.
> >
> > Please correct me if I made any misunderstanding or incorrect assumptions about this estimation. I look forward to the authors' response to these questions and concerns.

---

> > > ### Author Response · Authors · 2024-11-28
> > >
> > > **Revision of the Theorem 4.1**: We have revised Theorem 4.1 in the submitted manuscript to reflect the suggested changes. We explicitly define the mask and the sampling process alongside clearly stating that there exists a mask for which the results of Theorem 4.1 hold. Thank you for your continued feedback.
> > >
> > > **Masking**: In Theorem 4.1, we prove a bound constraining the outputs of pruned and unpruned LoRAs using a mask in $\Delta W \cdot U$ form. This mask is sampled from a Bernoulli distribution ($\mathcal{B}(\pi)$) where, pragmatically, the elements of the low-rank residual are *zeroed-out* with probability $\pi$. The direct implementation of this formulation without any changes would be inefficient since the exact same number of parameters compared to a fully parameterized low-rank residual would have to be computed just for the majority of these parameters to be zeroed-out using the mask. We instead modify the formulation of the low-rank residuals so the parameters that will be zeroed-out would not be computed from the start. This is done by zeroing out the rows of the $B$ matrix and columns of the $A$ matrix in a low-rank formulation (i.e. in LORAs) that results in $\Delta W = u_{row}B\;u_{col}A$. While it is true that matrix multiplication is non-commutative, this formulation is the exact same in terms of the final result compared to $\Delta W \cdot U$. This is because any element that is zeroed out in the original formulation will have a corresponding row and column that can be zeroed out from the decomposed formulation. Therefore, this act of splitting the mask into two components does not alter Theorem 4.1 in any way since it does not impact the results of masking and allows for obtaining the same exact sparsity ratio which is the main focus of the theorem.

---

> ### Author Response · Authors · 2024-11-28
>
> **Memory Reduction for LLAMA2-7B**:
> We greatly appreciate the reviewer’s detailed breakdown of LoRA’s memory footprint. We have revised the manuscript to report the exact trainable parameters across different methods. Below is this updated table:
> | Task   | FT            | LoRA           |  Masking          | Partial-LoRA          |
> |--------|---------------|----------------|-------------------|-----------------------|
> | SST-2  | 94.7(6.7B)    | 95.4(4.2M)     |  95.5(68K)        | 94.9(68K)             |
>
> We acknowledge the reviewer’s calculation, which accurately estimates that the memory overhead of a single LoRA adapter (e.g., 4.2M parameters with AdamW optimizer states) constitutes a small fraction of the total memory footprint for LLAMA2-7B. We bring here two scenarios where LoRA pruning for LLAMA2-7b is crucial to the application:
>
> **High-Rank Finetuning**: We consider task vectors [a] where the model is required to be fine-tuned on a small dataset with the strict requirement of a large rank for the adapter. Moreover, the application of the adapter is not only done on query (Q) and value (V) matrices but on every linear layer. In the case of LLAMA2-7b using LoRA with a rank of 256 on every linear layer, an unpruned LoRA would require 639 million parameters which would constitute just over 10GBs of memory. In such a case a pruned LoRA would be the only possible solution to obtain a task vector on that dataset.
>
> **Multi-LoRA**: Our initial claim was motivated by scenarios where multiple LoRAs are combined within a single model—a setting explored in prior works such as [b], [c], and [d]. As a future work, we hope to extend our masks to have potentially non-overlapping masks for different LoRAs. In [b], the approach involves training thousands of LoRA adapters and combining them with varying coefficients in a continual learning paradigm. In such cases, reducing the size of each LoRA adapter becomes critical. By sparsifying LoRA and reducing the trainable parameter count from 4.2M to just 68K, the model can accommodate a significantly larger number of adapters, enhancing scalability and flexibility in multi-task or continual learning applications.
>
> We appreciate the opportunity to clarify this point and will incorporate these insights into the revised manuscript. We hope this addresses the reviewer’s concerns and provides a more nuanced perspective on the benefits of sparsifying LoRA in practical use cases where the use case extends beyond a conventional experiment on a curated dataset. We hope that the reviewer considers raising their score for our paper and supporting its acceptance.
>
> [a] Ilharco, G., Ribeiro, M., Wortsman, M., Gururangan, S., Schmidt, L., Hajishirzi, H., & Farhadi, A. (2022). Editing Models with Task Arithmetic. ArXiv, abs/2212.04089.
>
> [b] Sheng, Y., Cao, S., Li, D., Hooper, C., Lee, N., Yang, S., Chou, C., Zhu, B., Zheng, L., Keutzer, K., Gonzalez, J.E., & Stoica, I. (2023). S-LoRA: Serving Thousands of Concurrent LoRA Adapters. ArXiv, abs/2311.03285.
>
> [c] Huang, C., Liu, Q., Lin, B., Pang, T., Du, C., & Lin, M. (2023). LoraHub: Efficient Cross-Task Generalization via Dynamic LoRA Composition. ArXiv, abs/2307.13269.
>
> [d] Zhao, J., Wang, T., Abid, W., Angus, G., Garg, A., Kinnison, J., Sherstinsky, A., Molino, P., Addair, T., & Rishi, D. (2024). LoRA Land: 310 Fine-tuned LLMs that Rival GPT-4, A Technical Report. ArXiv, abs/2405.00732.

---

> ### Comment · Reviewer_H8Ln · 2024-11-28
> **Missing link between direct formulation and decomposed formulation**
>
> > This is done by zeroing out the rows of the $B$ matrix and columns of the $A$ matrix in a low-rank formulation (i.e. in LORAs) that results in $\Delta W = (u_{row}B)(u_{col}A)$. While it is true that matrix multiplication is non-commutative, this formulation is the exact same in terms of the final result compared to $\Delta W \cdot U$.
>
> I appreciate the authors' clarification on this point. However, from my perspective, the above statement implies the following proposition holds:
>
> **(Existence of Decomposable Winning Tickets)** Given $\Delta W = BA$, there exists a mask $U$ such that
> 1. error caused by $U$ is constrained by the bound specified in Theorem 4.1,
> 2. there exist $u_{row}\sim\mathcal{B}(p_{row})$ and $u_{col}\sim\mathcal{B}(p_{col})$ such that $(u_{row}B)(u_{col}A) = \Delta W \cdot U$.
>
> Unless the above proposition holds, it is difficult to explain why the direct implementation can be *exact same* as a decomposed implementation, or why there always exists a decomposed implementation that is a winning ticket.
> Maybe I misunderstood the authors' point, and I would appreciate it if the authors could provide more insights on this.
>
> I delve deeply into this point because it is the critical link between the proposed method (decomposed formulation) and the proof of the lottery ticket hypothesis (Theorem 4.1, direct formulation). Missing this link would significantly weaken the paper's persuasiveness in drawing a connection between the low-rank adapter models and the lottery ticket hypothesis.

---

> > ### Author Response · Authors · 2024-11-29
> >
> > **Decomposition of the Mask**: The only constraint on $U$ is the sparsity ratio determined at each layer of the model that the LoRA is added onto, represented by $p^l$ in Theorem 4.1. This sparsity can be induced through $u_{row}$ and $u_{column}$. Given that the decomposed formulation can represent the same sparsity (albeit through different sparsity ratios on the decomposed formulation that leads to the same sparsity for $U$), Theorem 4.1 would hold. By "exact" we were referring to this process where the sparsity ratio will be exactly the same.
> >
> > To elaborate more on this from another perspective, we refer to [a], Lemma 2.4 (Subset sum approximation in ER Networks) that is the basis for one-for-one LTH theorems. Based on this Lemma:
> >
> > Assume $X_1, \cdots, X_n$ to be independent, uniformly distributed random variables so that $X_i \sim U([−1, 1])$ and $M_1, \cdots, M_n$ to be independent, Bernoulli distributed random variables so that $M_i \sim \mathcal{B}(p)$ for a $p \gt 0$. Then given variables $\epsilon, \delta \in (0, 1)$, for any $z \in [−1, 1]$ there exists a subset $I \subset [n]$ of the set $M$ so that with probability at least $1 − \delta$ we have $|z − \sum_{i\in I} M_iX_i| ≤ \epsilon$ if:
> >
> > $n \geq C \frac{1}{\log (1 /(1-p))} \log \left(\frac{1}{\min (\delta, \epsilon)}\right)$.
> >
> > Where $C$ is a generic constant derived in [b].
> >
> > We essentially use proxies to obtain $p^l$ and allow the training to find the subset that allows for approximation of the sum problem. In line with this, we clearly state in lines 226 to 232, that we use proxies to find sparsity ratios and masks that would enable this procedure with the reasoning that this mask is unknown beforehand since the target LoRA is unknown beforehand. Therefore, the theoretical framework in our work informs the practical implementation proposed in section 4.2 but has no way of directly providing access to the mask due to how the target model is unknown. We will add this explicitly to the paragraph of line 232.
> >
> > [a] Gadhikar, A., Mukherjee, S., & Burkholz, R. (2022). Why Random Pruning Is All We Need to Start Sparse. International Conference on Machine Learning.
> >
> > [b] Lueker, G.S. (1998). Exponentially small bounds on the expected optimum of the partition and subset sum problems. Random Struct. Algorithms, 12, 51-62.

---

> > > ### Author Response · Authors · 2024-11-29
> > >
> > > **Update on LLAMA2 results**:
> > > We provide here the updated results of LLAMA2-7b on the Copa and WSC as two additional datasets.
> > >
> > > | Task   | FT            | LoRA           |  Masking          | Partial-LoRA          |
> > > |--------|---------------|----------------|-------------------|-----------------------|
> > > | SST-2  | 94.7(6.7B)    | 95.4(4.2M)     |  95.5(0.068M)     | 94.9(0.068M)          |
> > > | WSC    | 70.2(6.7B)    | 70.2(4.2M)     |  64.52(0.7M)      | 66.4(0.7M)            |
> > > | Copa   | 87.0(6.7B)    | 85.0(4.2M)     |  88.0(0.068M)     | 84.0(0.060M)          |
> > >
> > > Here is a brief description of the two new tasks:
> > >  - WSC: The Winograd Schema Challenge (WSC) dataset tests pronoun resolution by providing sentences with ambiguous pronouns, where the task is to identify the correct referent using commonsense reasoning.
> > >  - Copa: The Choice of Plausible Alternatives (COPA) dataset consists of 1,000 premises, each with two possible alternatives, where the task is to determine which alternative is more plausible as a cause or effect of the premise.
> > >
> > > WSC Dataset Example:
> > >  - Sentence: The trophy doesn’t fit in the suitcase because it is too large.
> > >  - Question: What does "it" refer to?
> > >  - Answer: The trophy.
> > >
> > > COPA Dataset Example:
> > >  - Premise: The vase fell off the shelf.
> > >  - Choice 1: It was hit by a ball.
> > >  - Choice 2: It was on sale.
> > >  - Correct Answer: Choice 1 (cause-effect relationship).
> > >
> > > On the Copa dataset, our approach approximates LoRAs with even fewer number of parameters which is determined by our subnetwork extraction method alongside the 90% margin. On the WSC dataset, our subnetwork extraction asks for more trainable parameters compared to the other two tasks reflecting the difficulty of this commonsense reasoning task leading to a slight performance decrease with pruning. Although due to the adaptive nature of our approach, we do obtain a higher EMA compared to [a].
> > >
> > > [a] Xu, Jing, and Jingzhao Zhang. "Random Masking Finds Winning Tickets for Parameter Efficient Fine-tuning." arXiv preprint arXiv:2405.02596 (2024).

---

> > > > ### Author Response · Authors · 2024-12-02
> > > >
> > > > **Link between Theorem 4.1 and Impelementation**: We elaborate further on the efficient implementation of Theoreom 4.1 used in the paper. The reviewer suggested that for the link between our Theorem and implementation to hold, the decomposed masks have to represent $\mathbf{U}$ itself. However, based on the constraints on the mask in Theorem 4.1, emulation of the sparsity ratio by the decomposed masks is sufficient. We provide here the implication of such a suggestion:
> > > >
> > > > The masking without any decomposition is represented in the form $\Delta \mathbf{W} \circ \mathbf{U}$ (or equivelantly ($(\mathbf{B}\mathbf{A}) \circ \mathbf{U}$ for LoRA) where $\circ$ represents the Hadamard product. Our implementation uses the formulation $u_{r} \circ B$ and $u_{c} \circ \mathbf{A}$ where the rows of $\mathbf{B}$ are masked using $u_r$ and columns of $\mathbf{A}$ are masked using $u_c$. To avoid using Hadamard products for the decomposed formulation, we can represent the same formulation using diagonal variants of the $u_r$ and $u_c$ in the form of $\mathbf{U}_r$ and $\mathbf{U}_c$. Then, for each element in row $i$ and column $j$, we have,
> > > >
> > > > $M^\text{decomposed}\_{ij} = \mathbf{U\_r}\mathbf{B}(\mathbf{U\_c}\mathbf{A}^\top)^\top = \sum\_{k=1}^{d}(\mathbf{U}\_r)\_{ii}\mathbf{B}\_{ik}(\mathbf{U}\_c)\_{jj}\mathbf{A}\_{kj}\,,$
> > > >
> > > > and
> > > >
> > > > $M^\text{Masked}\_{ij} = (\mathbf{B}\mathbf{A})\_{ij} \circ \mathbf{U}\_{ij} = \sum\_{k=1}^{d} \mathbf{B}\_{ik} \mathbf{A}\_{kj} \mathbf{U}\_{ij}\,.$
> > > >
> > > >
> > > > Here, we aim to establish a connection between $\mathbb{E}[M^{\text{masked}}\_{ij}]$ and $\mathbb{E}[M^{\text{decomposed}}\_{ij}]$. Since the only variables in these equations are $\mathbf{U}, \mathbf{U}\_r, \mathbf{U}\_c$, then $\mathbb{E}[M^{\text{masked}}\_{ij}] = (\mathbf{B}\mathbf{A})\_{ij} \circ \mathbb{E}[\mathbf{U}\_{ij}]$ and $\mathbb{E}[M^{\text{decomposed}}\_{ij}]=\mathbb{E}[(\mathbf{U}\_r)\_{ii}]\(\mathbf{B}\mathbf{A}\)\_{ij}\mathbb{E}[(\mathbf{U}\_c^\top)\_{jj}]$.
> > > >
> > > > We note that for the two masks to have the same first moment (expectation), we have $\mathbb{E}[\mathbf{U}\_{ij}] = \mathbb{E}[(\mathbf{U}\_r)\_{ii}(\mathbf{U}\_c)\_{jj}] &nbsp; \forall &nbsp; i,j$ which would be the sufficient condition for satisfying the theorem.
> > > >
> > > > As clearly stated in revised version of Theorem 4.1, the mask $\mathbf{U}$ is sampled from a Bernoulli distibution with probability $p$. This is the only requirement for the mask where the expectation of this mask has to equal to $\mathbb{E}[\mathbf{U}_{ij}]=p$ (note that all elements are sampled independently). Assuming that we require a sparsity ratio of $p$, the RHS of the above equation has to now emulate this sparsity, that is
> > > > $\mathbb{E}[(\mathbf{U}\_r)\_{ij}]\mathbb{E}[(\mathbf{U}\_c)\_{ij}]=p\,.$
> > > >
> > > > Therefore, any sparsify ratio could be satisfied since we can control the way rows and columns are sampled where the decomposed masks are themselves sampled from the Bernoulli distribution for the Partial-LoRA variant.

---

> > > > > ### Comment · Reviewer_H8Ln · 2024-12-02
> > > > >
> > > > > I would express the great appreciation for the authors' efforts in addressing my concerns. In the latest response, the existence of $U$ satisfying the conditions of Theorem 4.1 is proven. However, what's about of the existence of diagonal $U_r$ and $U_c$, which satisfy some sparsity ratio, such that
> > > > > $$
> > > > > \max_{x\in D}\|f_T(x;\Delta W_T) - f_{LoRA}(x;U_r B A U_c^T)\| \le \epsilon
> > > > > $$
> > > > >
> > > > > If there is any theoretical evidence supporting the existence of such $U_r$ and $U_c$, it would be a significant contribution to the paper. I look forward to the authors' response to this question.

---

> > > > > > ### Author Response · Authors · 2024-12-03
> > > > > >
> > > > > > Thank you for your continued responses and valuable feedback. We'd like to clarify that for our approach the existance of $\mathbf{U}_r, \mathbf{U}_c$ is *not* necessary. The existence of $\mathbf{U}$ is already proven in Theorem 4.1 and our comments previous to that. We merely suggested an efficient implementation of the theorem.
> > > > > >
> > > > > > The existence of specific diagonal matrices $\mathbf{U}_r$ and $\mathbf{U}_c$ in relation to $\mathbf{U}$ is not relevant for the bound in Theorem 4.1. What matters is the sparsity ratio that can be induced on the LoRA residuals. As shown in our previous comment, both masking approaches ($\mathbf{W}^{\text{masked}}=\Delta \mathbf{W} \circ \mathbf{U}=(\mathbf{BA}) \circ \mathbf{U}$ and $\mathbf{W}^{\text{masked}}=\mathbf{U}_r \mathbf{B} (\mathbf{U}_c \mathbf{A}^\top)^\top$), in our previous comment, are capable of inducing the same sparsity ratio $p$, which is the only constraint required by Theorem 4.1 in our paper, original construction of LTH in Theorem 2.5 in [a] and the subset sum approximation proof in Lemma A.1 in [a]. Therefore, *the bound mentioned by the reviewer remains intact due to the matching of the sparsity ratio regardless of the decomposition and masking implementation*. Note that we, initially, used the non-decomposed formulation and upon switching to the decomposed implementation, we observed the same performance in our vision experiments in Figure 2. Our implementation is more efficient on a GPU.
> > > > > >
> > > > > > We're happy to elaborate further and hope this additional context will lead the reviewer to reconsider our score.
> > > > > >
> > > > > > [a] Gadhikar, A., Mukherjee, S., & Burkholz, R. (2022). Why Random Pruning Is All We Need to Start Sparse. International Conference on Machine Learning.

---

### Official Review · Reviewer_Pp44 · 2024-11-04

**Soundness:** 3
**Presentation:** 4
**Contribution:** 3
**Rating:** 6
**Confidence:** 3

**Summary:**

This paper extends the concept of winning tickets to low-rank adapters, demonstrating through theoretical and experimental evidence that masking a significant proportion of LoRA adapters can be done without compromising performance. The authors introduce Partial-LoRA, which strategically selects sparse low-rank parameters, achieving substantial reductions in trainable parameters (up to 87%) while maintaining or even enhancing model performance in certain cases.

**Strengths:**

1. This paper is well-written and interesting. It introduces the concept of winning tickets to low-rank adapters in a novel way and is backed by theoretical grounding.
2. From an empirical perspective, the method of masking a proportion of parameters offers more flexibility compared to existing approaches and shows strong performance across multiple experiments on vision and language models.

**Weaknesses:**

1. The datasets chosen by the authors are relatively simple, and the model used is comparatively small. In such tasks, which inherently may require fewer parameters[1] to learn effectively, the performance gap between full fine-tuning and PEFT methods, including LoRA, tends to be minimal. In more challenging tasks where LoRA underperforms compared to full fine-tuning possibly due to capacity limitations[2], it is unclear whether Partial-LoRA would maintain its advantage.
2. A discussion on the impact of the few-shot dataset for the sparsity ratio is necessary. While larger datasets could provide more accurate estimates, they would also require significantly more time to compute.

[1] Aghajanyan, Armen, Luke Zettlemoyer, and Sonal Gupta. "Intrinsic dimensionality explains the effectiveness of language model fine-tuning." *arXiv preprint arXiv:2012.13255* (2020).
[2] Biderman, Dan, et al. "Lora learns less and forgets less." *arXiv preprint arXiv:2405.09673* (2024).

**Questions:**

1. Figure 2 shows that Partial-LoRA, with its sparsity ratio derived from the 90% performance threshold, is comparable to or outperforms LoRA. Would it be possible to set this threshold lower to achieve a higher sparsity ratio? What is the maximum sparsity ratio at which performance remains stable?
2. To reduce memory usage, two strategies could be considered: (a) lowering the rank of LoRA or (b) increasing the sparsity of LoRA. Does the author suggest that using a higher rank with a more sparse LoRA could improve performance, as implied by Section 5.5 and Figure 6?

---

> ### Author Response · Authors · 2024-11-25
>
> Thank you for your thorough and constructive review of the paper and for recognizing our work as well-written and interesting.
>
> - **W1 - Dataset and model choices**: We use ViT-B-16 for vision and Deberta-v3-base for language models which are the commonly used benchmarking models for works on LoRAs [a, b, c] to be able to directly compare our work to the literature. Interestingly, smaller models can be harder to prune [d]. We are running experiments on LLAMA2-7b as a larger model and will report the results when completed.
>
> - **W1 - Case where LORA underperforms full-finetuning**: Our method highlights the lottery ticket hypothesis in the low-rank approximations based on which we propose Partial-LORA. Therefore, we don't expect Partial-LoRAs outperforming the full finetuning in the case where LoRAs underperform. Nevertheless, we report here the results of full finetuning on GTSRB and CIFAR10 datasets where full finetuning performs better than parameter efficient low-rank counterparts alongside each method's paramter count in parentheses. In both cases our method exhibits marginal difference compared to LORA while utilizing fraction of parameters. We will add the full finetuning performance results to the paper.
>
> **Table I: Comparison to full finetuning on CIFAR-10 and GTSRB datasets.**
> | Method       | CIFAR-10 | GTSRB  |
> |--------------|----------|--------|
> | FT           | 98.50 (86M)   | 99.20 (86M)   |
> | Partial-LoRA | 94.21 (0.17M) | 94.72 (0.26M) |
> | LoRA         | 94.50 (0.67M) | 95.04 (0.67M) |
>
> - **W2 - Few-shot and Sparsity Ratio** As shown in Figure 13 in Appendix E, over different number of shots, there is a large overlap between the extracted subnetwork from both SVD-based and SNIP (gradient-based) methods. Therefore, considering that one of the main messages of our work is about the importance of sparsity ratios, the insignificant changes in the top indices across different shots shows that the sparsity ratio is not dependent on the variability of the shots. We'll add a discussion on this topic to the main text of the paper. On the matter of time it takes for the computation of the sparsity ratio, if gradient-based methods such as SNIP or IMP are used (Section 4.2), larger datasets may need more time and memory to extract the ratios. However, using our proposed SVD-based approach, due to the forward-only aspect of our method, the timing does not change significantly when extracting subnetworks for larger datasets. As an example, for the case of larger models such as LLAMA2-7b, the subnetwork extraction process takes less than an hour on a 4090 GPU.
>
> - **Q1 - Training Threshold**: That is an interesting experiment considering that this threshold directly determines the resulting sparsity ratio. The choice of 90% accuracy threshold is adopted from the intrinsic dimensionality literature [e] which themselves take that threshold from [f] where the 90% accuracy margin is deemed a satisfactory condition. We will complement our experiments with 70% and 80% margins.
>
> - **Q2 - Different Method of Memory Usage Reduction**: Great observation! We suggest using higher-rank because our results show that only a small number of parameters are required to finetune the model, i.e., Sparse residuals can make the necessary adjustments to access critical subspaces. By increasing the rank of the residuals, we can more precisely modify the top subspaces of the model while still using fewer parameters than a fully-parameterized LoRA. Meanwhile, simply increasing the number of parameters may not lead to better performance since these critical subspaces are already accessible. That said, there is a limit: reducing the number of parameters too much can prevent access to these important subspaces, hurting performance. We'll expand on this idea in the discussion section.
>
> [a] Hayou, S., Ghosh, N., & Yu, B. (2024). LoRA+: Efficient Low Rank Adaptation of Large Models. ArXiv, abs/2402.12354.
>
> [b] Zhang, Qingru et al. “Adaptive Budget Allocation for Parameter-Efficient Fine-Tuning.” ArXiv abs/2303.10512 (2023): n. pag.
>
> [c] Hu, J.E., Shen, Y., Wallis, P., Allen-Zhu, Z., Li, Y., Wang, S., & Chen, W. (2021). LoRA: Low-Rank Adaptation of Large Language Models. ArXiv, abs/2106.09685.
>
> [d] Xu, Jing, and Jingzhao Zhang. "Random Masking Finds Winning Tickets for Parameter Efficient Fine-tuning." arXiv preprint arXiv:2405.02596 (2024).
>
> [e] Aghajanyan, Armen, Luke Zettlemoyer, and Sonal Gupta. "Intrinsic dimensionality explains the effectiveness of language model fine-tuning." arXiv preprint arXiv:2012.13255 (2020).
>
> [f] Chunyuan Li, Heerad Farkhoor, Rosanne Liu, and Jason Yosinski. Measuring the intrinsic dimension
> of objective landscapes. arXiv preprint arXiv:1804.08838, 2018.

---

> > ### Author Response · Authors · 2024-11-26
> >
> > **Dataset and model choices**: We provide the results on LLAMA2-7b here and have added a new section in the revised version of the main text.
> > | Task   | FT            | LoRA           |  Masking (0.001%) | Partial-LoRA (0.001%) |
> > |--------|---------------|----------------|-------------------|-----------------------|
> > | SST-2  | 94.7(1e-6)    | 95.4(1e-4)     |  95.5(5e-2)       | 94.9(5e-2)            |
> >
> > Here we compare the results of our work against full fine-tuning, low-rank adaptation, and Balanced Masking techniques [e]. We have added this new work as a comparison to other masking techniques for PEFT approaches recommended by reviewer vQUE. The related work section of the main text has also been revised accordingly. As evident in the table above, for LLAMA2-7b on SST-2, the sparsification of LoRA is also possible with larger models. We will be adding more datasets to this table in the coming few days. Compared to Balanced Masking, we obtain similar performance without the requirement of an expensive grid search over the sparsity ratio since we use the sparsity ratio derivation technique proposed in Section 4.2 of our paper.
> >
> > We briefly bring a comparison to [a] here that can also be found in our related work section. Our approach employs an efficient masking technique that is non-ad-hoc with respect to how the residuals are masked. We theoretically justify our method using the literature on LTH where [a] does not. Perhaps more importantly, we employ a forward-only mechanism adaptive to the required capacity by the task and derive the required sparsity used for sampling our random masks. [a] does not provide any approach to deriving the probability of the Bernoulli distribution for the random masks and for experimentation, performs a grid search on the sparsity ratio as a hyperparameter.
> >
> > These results have also been added to the revised manuscript in Section 5.4 and the new Table 1. Additionally, we are prioritizing the LLM tests and will add the results with different margins and apply revisions with our proposal for Q2 shortly.
> >
> > [a] Xu, Jing, and Jingzhao Zhang. "Random Masking Finds Winning Tickets for Parameter Efficient Fine-tuning." arXiv preprint arXiv:2405.02596 (2024).

---

> > > ### Author Response · Authors · 2024-11-28
> > >
> > > **Sweeping margins**: Here, we present an analysis of the performance of Partial-LoRAs under varying accuracy thresholds, specifically adjusting the 90% margin to 70% and 80%. The evaluation was conducted on the Flowers and GTSRB datasets. The Flowers dataset was selected due to its higher performance compared to unpruned LoRA, while GTSRB was chosen as it demonstrated superior performance compared to VeRA [a] in our paper. The results are summarized in the table below:
> > >
> > > | Dataset  | 70%          | 80%          | 90%          |
> > > |----------|--------------|--------------|--------------|
> > > | GTSRB    | 93.11 (0.14M)| 93.61 (0.16M)| 94.72 (0.26M)|
> > > | Flowers  | 94.32 (0.11M)| 94.05 (0.17M)| 95.21 (0.32M)|
> > >
> > > On GTSRB, we still maintain the performance gap relative to VeRA with the reduction in the number of parameters while achieving similar parameter count to VeRA. On Flowers, the accuracy drops a percentage point when lowering the 90% threshold. This shows how suitable the 90% margin is in striking a balance between parameterization and performance. We will present these results as a new Appendix section with a visualization for clearer representation.

---

> > > > ### Author Response · Authors · 2024-11-29
> > > >
> > > > **Update on LLAMA2 results**:
> > > > We provide here the updated results of LLAMA2-7b on the Copa and WSC as two additional datasets.
> > > >
> > > > | Task   | FT            | LoRA           |  Masking          | Partial-LoRA          |
> > > > |--------|---------------|----------------|-------------------|-----------------------|
> > > > | SST-2  | 94.7(6.7B)    | 95.4(4.2M)     |  95.5(0.068M)     | 94.9(0.068M)          |
> > > > | WSC    | 70.2(6.7B)    | 70.2(4.2M)     |  64.52(0.7M)      | 66.4(0.7M)            |
> > > > | Copa   | 87.0(6.7B)    | 85.0(4.2M)     |  88.0(0.068M)     | 84.0(0.060M)          |
> > > >
> > > > Here is a brief description of the two new tasks:
> > > >  - WSC: The Winograd Schema Challenge (WSC) dataset tests pronoun resolution by providing sentences with ambiguous pronouns, where the task is to identify the correct referent using commonsense reasoning.
> > > >  - Copa: The Choice of Plausible Alternatives (COPA) dataset consists of 1,000 premises, each with two possible alternatives, where the task is to determine which alternative is more plausible as a cause or effect of the premise.
> > > >
> > > > WSC Dataset Example:
> > > >  - Sentence: The trophy doesn’t fit in the suitcase because it is too large.
> > > >  - Question: What does "it" refer to?
> > > >  - Answer: The trophy.
> > > >
> > > > COPA Dataset Example:
> > > >  - Premise: The vase fell off the shelf.
> > > >  - Choice 1: It was hit by a ball.
> > > >  - Choice 2: It was on sale.
> > > >  - Correct Answer: Choice 1 (cause-effect relationship).
> > > >
> > > > On the Copa dataset, our approach approximates LoRAs with even fewer number of parameters which is determined by our subnetwork extraction method alongside the 90% margin. On the WSC dataset, our subnetwork extraction asks for more trainable parameters compared to the other two tasks reflecting the difficulty of this commonsense reasoning task leading to a slight performance decrease with pruning. Although due to the adaptive nature of our approach, we do obtain a higher EMA compared to [a].
> > > >
> > > > [a] Xu, Jing, and Jingzhao Zhang. "Random Masking Finds Winning Tickets for Parameter Efficient Fine-tuning." arXiv preprint arXiv:2405.02596 (2024).

---

> > > > > ### Comment · Reviewer_Pp44 · 2024-12-03
> > > > > **Thank you for author's response**
> > > > >
> > > > > Thank you for the authors’ detailed reply and the newly added experiments! I now have a deeper understanding of the properties of Partial-LoRA, which I find to be a highly interesting and intuitive approach. I completely understand the constraints of time and computational resources during the rebuttal phase. However, the point I aimed to emphasize in W1 was about the "complexity of datasets" rather than the "complexity of models." For simple tasks like SST-2, models such as Llama2 can achieve satisfactory performance even with In-Context Learning. For such tasks, the parameter efficiency of Partial-LoRA is indeed very natural.
> > > > >
> > > > > However, for tasks that are more representative of real-world applications of large model fine-tuning—such as General SFT, Math, Code, or Knowledge Injection through Continued Pretraining(see evaluation in [1][2][3])—these often demand extensive training and parameter space. I'm still unsure whether Partial-LoRA can perform well in these scenarios.
> > > > >
> > > > > I agree with the authors' statement: "We use ViT-B-16 for vision and Deberta-v3-base for language models which are the commonly used benchmarking models for works on LoRAs [a, b, c] to be able to directly compare our work to the literature." However, I believe that demonstrating Partial-LoRA’s performance on tasks where fine-tuning is extensively used in practice would make the method more widely adopted and bring greater benefit to the community.
> > > > >
> > > > > [1]Wang, Shaowen, Linxi Yu, and Jian Li. "LoRA-GA: Low-Rank Adaptation with Gradient Approximation." arXiv preprint arXiv:2407.05000 (2024).
> > > > >
> > > > > [2]Jiang, Ting, et al. "MoRA: High-Rank Updating for Parameter-Efficient Fine-Tuning." arXiv preprint arXiv:2405.12130 (2024).
> > > > >
> > > > > [3]Biderman, Dan, et al. "Lora learns less and forgets less." arXiv preprint arXiv:2405.09673 (2024).

---

> > > > > > ### Author Response · Authors · 2024-12-03
> > > > > >
> > > > > > Thank you for the clarification and helpful insights. Other than SST-2, the other two datasets, commonsense reasoning [a] (WSC dataset) and cause-effect relationships [b,c] (Copa) we use for evaluating LLAMA2-7b are known for their difficulty and complexity due to the need for dealing with nuanced problems. Solutions to these problems require capabilities that mirror the patterns needed to address real-world challenges. Moreover, testing on these datasets allows us to compare our work to that of the SotA on low-rank adaptation in a fair scenario.
> > > > > >
> > > > > > Additionally, we note that the variants of LoRA such as MoRA or LoRA-GA are designed to improve LoRAs from specific perspectives such as increasing the rank or gradient alignment. Even then, these modifications are still not incompatible with our proposed masking approach. Since our approach was aimed at approximating LoRAs and extending the LTH theoretical framework to low-rank adaptation, we chose LoRA to avoid the over-complication of the theoretical framework. However, we will add this interesting trajectory on the effects of Partial-LoRA on specialized variants of LoRA as a future work to our conclusion section.
> > > > > >
> > > > > > [a] Zhang, H., Zhao, X., & Song, Y. (2020). WinoWhy: A Deep Diagnosis of Essential Commonsense Knowledge for Answering Winograd Schema Challenge. Annual Meeting of the Association for Computational Linguistics.
> > > > > >
> > > > > > [b] Kavumba, P., Inoue, N., Heinzerling, B., Singh, K., Reisert, P., & Inui, K. (2019). When Choosing Plausible Alternatives, Clever Hans can be Clever. ArXiv, abs/1911.00225.
> > > > > >
> > > > > > [c] Yuan, M., Whitehouse, C., Chamoun, E., Aly, R., & Vlachos, A. (2024). PRobELM: Plausibility Ranking Evaluation for Language Models. ArXiv, abs/2404.03818.

---

### Meta-Review · Area_Chair_T7yW · 2024-12-21

**Metareview:**

The paper investigates the application of the lottery ticket hypothesis to LoRA and introduces Partial-LoRA, a method for sparse parameter tuning using random masking. The approach aims to reduce the number of trainable parameters while maintaining or improving performance. While reviewers acknowledged the novelty of extending LTH to LoRA and appreciated the strong empirical results on vision and language tasks, they raised concerns about the limited evaluation of larger models, insufficient exploration of complex tasks, and the need for theoretical justification. Despite revisions during the rebuttal period, these issues remained only partially resolved. I recommend rejection.

**Additional Comments On Reviewer Discussion:**

Reviewers raised concerns about the limited evaluation of larger models, insufficient exploration of complex tasks, and the need for theoretical justification. The authors addressed these points by providing additional experiments and clarifications, but the responses were not sufficient to fully resolve the concerns.

---

### Decision · Program_Chairs · 2025-01-22

Reject